# An experimentally validated approach to automated biological evidence generation in drug discovery using knowledge graphs

Saatviga Sudhahar [1] ✉, Bugra Ozer [1], Jiakang Chang[1], Wayne Chadwick [1], Daniel O'Donovan [1], Aoife Campbell[1], Emma Tulip[1], Neil Thompson[1] & Ian Roberts [1]

Explaining predictions for drug repositioning with biological knowledge graphs is a challenging problem. Graph completion methods using symbolic reasoning predict drug treatments and associated rules to generate evidence representing the therapeutic basis of the drug. Yet the vast amounts of generated paths that are biologically irrelevant or not mechanistically meaningful within the context of disease biology can limit utility. We use a reinforcement learning based knowledge graph completion model combined with an automatic filtering approach that produces the most relevant rules and biological paths explaining the predicted drug's therapeutic connection to the disease. In this work we validate the approach against preclinical experimental data for Fragile X syndrome demonstrating strong correlation between automatically extracted paths and experimentally derived transcriptional changes of selected genes and pathways of drug predictions Sulindac and Ibudilast. Additionally, we show it reduces the number of generated paths in two case studies, 85% for Cystic fibrosis and 95% for Parkinson's disease.

Discovering safe and effective treatments for rare diseases presents a formidable challenge, starting with the sourcing, normalising, and integration of copious, diffuse and diverse data sources that inform drug discovery. When it comes to the more than 7000 rare and genetic disorders, valuable information is frequently dispersed across various databases, encompassing clinical symptoms, impacted pathways, animal models, and potential treatments. To address this issue, AI-driven computational tools and knowledge can be harnessed to interconnect this diverse data, enabling the prediction of innovative drug candidates. Typically, existing computational methods generate an overwhelming number of therapeutic hypotheses, necessitating labour-intensive manual curation by experts specializing in the respective disease. This process involves a significant amount of time dedicated to establishing the therapeutic linkage between the drug and the disease, given the identification of the mechanism of action is pivotal in establishing clinical tractability.

Knowledge graphs (KG) have been used extensively in the recent past to solve complex problems in life sciences including drug discovery for rare diseases. Knowledge graphs are constructed with head entity-relation-tail entity (h, r, t) triples where entities correspond to nodes and relations correspond to links connecting the entities. Biological knowledge graphs are constructed with biological nodes such as drugs, diseases, genes, pathways, phenotypes, proteins etc and the links between these nodes. Knowledge base completion (KBC)[1] is the task of predicting the tail entity $t$, given the head entity $h$, and the relation $r$, or head entity $h$ given the tail entity $t$ and relation $r$. It can also be used to predict unseen relations between the head and tail entities. Several approaches have been proposed in the past for KBC that learn a continuous vector space representation for entities and relations. These methods include translational models using distance-based scoring (TransE[2], TransH[3], RotatE[4]), semantic matching models (RESCAL[5], DistMULT[6], ComplEX[7]), Graph convolutional networks

[1]Healx Ltd, Cambridge, United Kingdom. ✉e-mail: saatviga.sudhahar@healx.io

(GCN[8], R-GCN[9]), Attention networks (GAT[10]) and context-based encoding approaches (KG-BERT[11]). Yet these models cannot produce a rationale for the predictions. Alternatively, there are comparatively few models such as AnyBURL[12,13], Minerva[14] that target the symbolic space capable of also producing a set of learnt logical rules for each prediction. Many recent studies have focused on the application of existing drugs for new therapeutic indications using biological knowledge graphs, for example, by identifying potential candidates based on shared biological mechanisms or targets. One of those has built a billion-edge biomedical KG from millions of biomedical documents to identify potential drug repositioning candidates for diseases with unknown treatment[15]. Biological KGs have been built integrating preclinical, clinical, and literature evidence from public sources to predict and rank genes leading to potential mechanisms of EGFRi resistance[16]. To inform clinical decision-making, knowledge graphs have been built from experimental data, public databases, and literature to augment and enrich proteomics data[17]. Few other studies have used knowledge graph completion models for drug repositioning in cancer and auto-immune conditions[16,18,19].

The list of ranked predicted drugs by KBC models can differ across models even with the same drug and disease data due to differences in model logic. This is observed usually with different ranking evaluation scores for different models[20]. Notwithstanding, these algorithms generate prediction lists of hundreds of drugs with the potential to treat the query disease. Real world limitations including cost, assay availability and capacity, mean that only a limited number of drug predictions can be experimentally tested preclinically for each disease. This constraint necessitates drug discovery scientists to devise methods for filtering drug predictions, prioritizing those that have a higher likelihood of demonstrating efficacy and safety in preclinical experiments. There are various strategies to increase success, but it all comes down to understanding at a biological level how a proposed treatment might be therapeutically beneficial for the patient. At its simplest level, this reduces to establishing an overlap between mechanisms of disease biology with the biology that the drug targets. We call this process establishing "therapeutic rationale", a comprehensive rationale providing a full picture of how a drug may be useful for treating the patient. It considers the current complete understanding of disease biology, including known causal genes and perturbed biological pathways and is established by combining several techniques including extraction of "evidence chains", a set of paths in the knowledge graph that explain the relationships between a disease and a target drug, connected via biological entities.

The explainability of predictions made by a model is addressed by relatively few methods, generally based on logical and path-based approaches that are capable of providing the user with an explicit explanation path that may serve as a justification for experimentally testing the prediction. One of the pioneers is the Path Ranking Algorithm (PRA)[21,22]. KG embeddings learnt from a KBC model have been used to create novel ranked paths between drugs and diseases, but the approach was limited to one-hop explainable paths[23]. Reasoning with logical rules has been addressed in areas such as Markov logic networks (MLNs)[24], however, such techniques typically do not scale well to modern, large-scale KGs. Recently, symbolic models such as RuleN[25], its successors AnyBURL[12,13], MINERVA[14] and PoLo[26] mine logical rules using a reinforced search-based path sampling for knowledge graph reasoning considering it as a neural-driven multi-hop problem to make predictions. Rules learnt during the learning phase are probabilistically annotated with confidence scores that represent the probability of predicting a correct fact with the rule. More details are provided in "Methods". The following example shows a rule explaining how compound $X$ can treat disease $Y$.

compound_disease_treats($X,Y$) $\Leftarrow$ compound_gene_binds($X,A$),
compound_gene_activates($B,A$), compound_in_trial_for($B,Y$)

Compound $X$ binds to gene $A$, which is activated by compound $B$, which is in trial for the disease $Y$.

The rule can be written in a simplified form as follows.

Compound$-binds \rightarrow Gene \leftarrow activates-$Compound$-in\,trial\,for \rightarrow$ Disease

Given a compound prediction Lumacaftor for the disease cystic fibrosis and the above rule supporting the prediction we generate a path or an evidence chain from the Healx KG as shown in Fig. 1a. The Healx KG is built from several publicly available and proprietary data sources and internally curated data. Details of nodes and edge types in this KG and their sources are provided in "Methods" under "Data".

Several other examples of rules and the corresponding paths that could be generated from the graph are shown in Fig. 1b. They all show the therapeutic basis of how a drug is connected to the disease. A limitation of this approach is the vast amount of evidence chains that can be generated for a single drug prediction that is infeasible to review by human experts in a reasonable time. The number of evidence chains grow with the number of rules associated with each prediction. Notwithstanding, in the biological context of the disease of interest many of the rules associated with predictions are irrelevant, redundant, or not beneficial to establishing a molecular understanding of efficacy or safety (hereafter, uninformative). Evidence chains of this class do not inform on therapeutic action, efficacy, or safety. Neither do they enable decision-making in a drug discovery programme and can be safely discarded. For example, the evidence generated for the drug Tobramycin as shown in Fig. 1c contains two ancestor relationships related to cystic fibrosis, one of them being "Rare genetic disease" and a "treats" relation with a very common condition "Cataract" which makes this chain uninformative when attempting to understand a potential cystic fibrosis treatment. The disease-disease ancestor relationships are derived from MONDO[27] and Orphanet[28] ontologies.

Rules generated for each prediction may also differ according to the disease of interest. Therefore, it's not possible to construct a universal set of rules to be used in all disease case studies since a new disease study might involve a new rule that has not been previously observed, yet still be biologically relevant. Manual curation is one solution to choosing the most appropriate rules required prior to generating evidence chains but would limit the usefulness of the approach for routine drug discovery given the inherent lack of scalability of expert human curation. In addition, manual curation poses a risk of personal bias that may skew the analysis and limit its reproducibility. To make evidence chains more relevant to the disease of interest, the disease landscape should be defined, and an automated way of generating biologically meaningful evidence should be implemented as we suggest here.

We start from a symbolic KBC model with a reinforcement learning-based reasoning approach AnyBURL[12,13] to make predictions and generate rules that support those predictions. We introduce a multi-stage pipeline that incorporates an automatic filtering model. This pipeline works in conjunction with AnyBURL's rules to filter and generate biologically relevant and meaningful explanations for the predictions it produces. A similar approach to ours has been used in previous work[29] where neural-driven RL based symbolic reasoning methods MINERVA[14] and PoLo[26] have been used to make biomedical KG predictions and reasoning chains applying pruning to the correct target type. The high-level ranked rules as given by MINERVA in the method appeared to be relevant but the low-level reasoning chains had to be manually reviewed by subject matter experts to confirm validity. Our primary contribution in this study is the automated extraction of pertinent high-level rules and biologically meaningful low-level reasoning chains feasible enough to be reviewed by experts. The inputs to the method include the knowledge graph and a list of genes and pathways of importance specific to the disease produced by curators and bioinformaticians in the disease landscape analysis

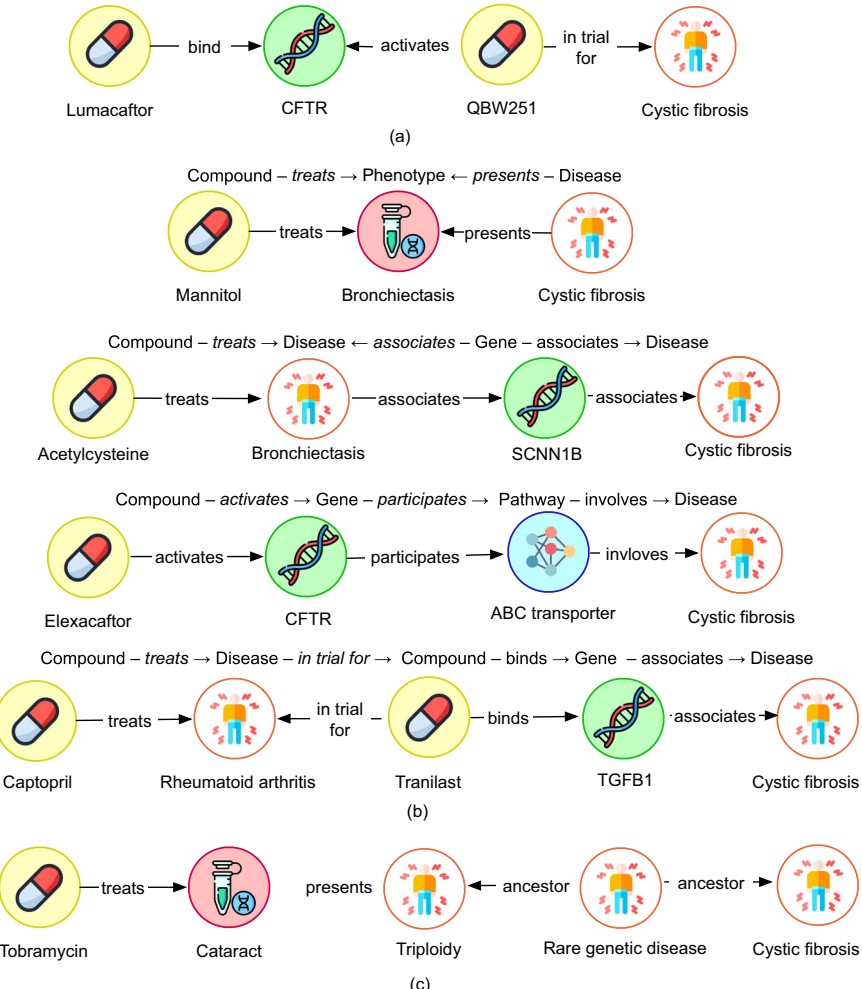

**Fig. 1 | Example rules and corresponding evidence chains generated from Healx KG. a** An example path from the graph satisfying the rule (Compound – binds → Gene ← activates – Compound – in trial for → Disease) for a given candidate drug prediction "Lumacaftor" in cystic fibrosis. **b** Example rules learnt by the model and paths generated for each rule. **c** Example of an uninformative evidence chain containing two ancestor relationships. Yellow symbol indicates the type compound, white indicates disease, green indicates gene, red indicates phenotype and blue indicates pathway.

(see "Methods"). The auto-filtering model includes a rule filter, significant path filter and a gene/pathway filter to subset only the biologically significant chains, reducing the total amount of evidence generated that would require human expert review. It also includes a deductive path builder that builds additional evidence chains using prioritized edge types when we have multiple chains involving the same nodes. More details are provided in "Methods", and a workflow diagram of the entire pipeline is presented in Fig. 2 in "Results".

We demonstrate our approach in Fragile X Syndrome (FXS), a rare genetic disorder where we validate the method against preclinical experimental data that showed a strong correlation between our evidence chains analysis results and experimentally derived transcriptional changes of selected genes and hallmark pathways of the predicted drug treatments Sulindac and Ibudilast. We also systematically tested the approach on two disease case studies cystic fibrosis and Parkinson's disease in order to validate the method against known treatments, assessing the general applicability of the approach while also covering different therapeutic areas. We chose these two diseases by, firstly, selecting for diseases that had at least one known approved treatment, diseases which were not excessively complex and had a genetic cause. We avoided complex and common diseases such as cancers and auto-immune conditions since validating the output against known information for these diseases would involve too many

genes and pathways and likely form an overly complex case study of our automatic evidence chains filtering approach. Cystic fibrosis met our selection criteria of having a causative gene (*CFTR*) with known treatments and is well-suited for validation purposes as we can verify whether the gene and treatment links are extracted by our pipeline. To further evaluate our approach, we chose Parkinson's disease, which is a more common disease with approved treatments for some of its phenotypes. Knowledge graph approaches have been applied to Parkinson's disease in recent studies using it for drug repositioning[30] and validating predicted targets against a gold standard dataset of genes associated with the disease[31].

This study represents a systematic evaluation of validating the evidence chains generated for diseases for therapeutic rationale using a knowledge graph completion model, path generation, and the application of an automatic filtering model. Consequently, the primary challenge in this work stemmed from the absence of any pre-existing benchmark data for quantitative validation of the approach. Creating a gold standard validation set of evidence chains for drug-disease pairs is prohibitively infeasible. It would require decision-making on what is the rationale for choosing the top-ranking evidence chains for any given drug-disease pair, differs for each disease of interest, and therefore would require a prohibitively large amount of time to conduct expert review from curators, pharmacologists and

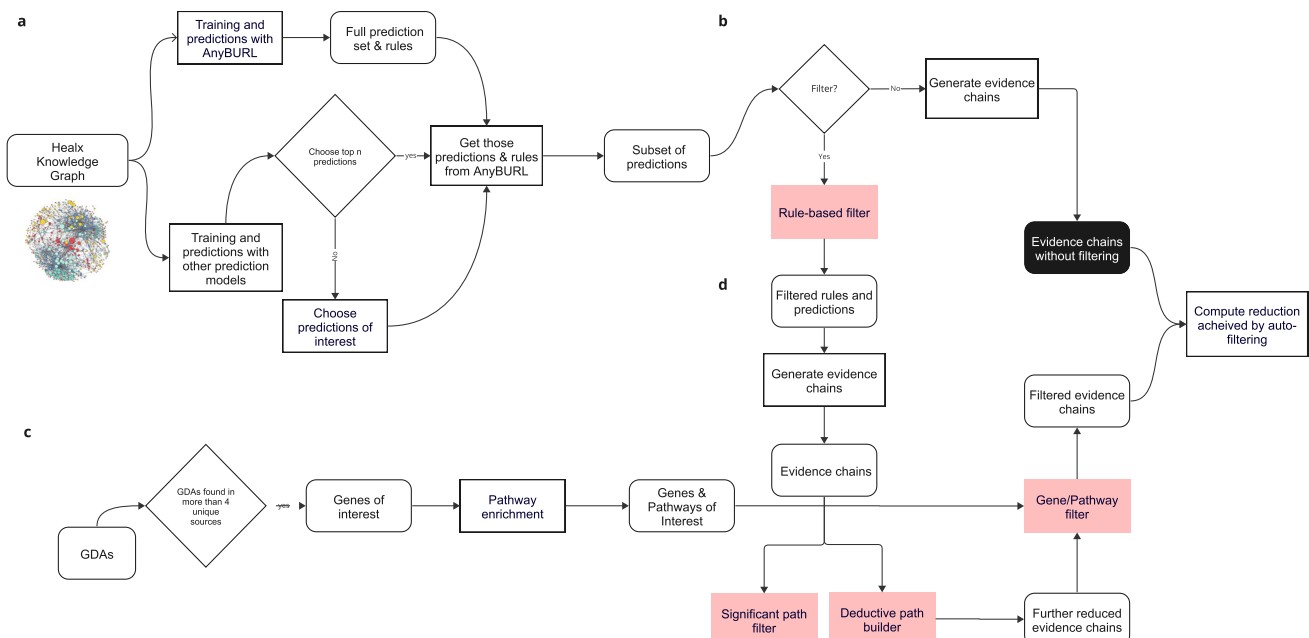

**Fig. 2 | Evidence chains generation workflow. a** Generating predictions and the corresponding rules from prediction models and AnyBURL **b** Generating evidence chains without the auto-filtering model. **c** Gathering biologically relevant gene-disease associations (GDAs) and pathways as input to the gene/pathway filter.

**d** Generating evidence chains with application of the auto-filtering model consisting of the rule-based, significant path, gene/pathway filters and the deductive path builder.

bioinformaticians. Therefore, we adopted a qualitative approach to address this challenge until such a gold standard validation set becomes available. We reached out to experts to evaluate whether the evidence chains generated by our case study diseases are relevant to the disease of interest and analysed if the disease related genes and pathways provided by curators are automatically extracted by our pipeline. In addition, and as an alternative to qualitative analysis, we explored ways to evaluate the efficiency of the auto-filtering method applied to rules and evidence chains. From the complete list of initial rules generated by AnyBURL for both cystic fibrosis and Parkinson's disease predictions, we manually curated a set of rules with the help of drug discovery scientists, which they thought could produce biologically relevant evidence chains. We made the entire set of rules from AnyBURL available for review and requested curators to assess the biological relevance of these rules in the context of any disease in general. This approach eliminated bias toward a specific disease and ensured comparability in the curation process. We checked whether the automated filtering method can retain all the curated rules before generating evidence. Lastly, we performed a case study on Fragile X syndrome, where we validated our approach against preclinical experimental data that demonstrated a strong correlation between our evidence chains analysis results and experimentally derived transcriptional changes of selected genes and hallmark pathways of the predicted drug treatments Sulindac and Ibudilast.

## Results
### Evidence chain generation workflow
Figure 2 shows a complete workflow diagram of the entire pipeline. It begins with predictions and proceeds through therapeutic hypothesis generation, gathering biological relevance, and generating evidence chains. The figure provides a comparison between the results obtained with and without automatic filtering, demonstrating the impact of this filtering process on the evidence chains.

**Predictions and rule generation.** We start with the input data, coming from the Healx KG as shown in Fig. 2a. AnyBURL and a set of other drug

prediction models are trained on this data and predictions are generated. AnyBURL produces a set of rules for each prediction. Either the top $n$ ($n = 100$ in our case) predictions from the prediction models or specific predictions of interest from the top $n$ are chosen for hypothesis generation. The criteria for choosing query predictions will differ according to therapeutic programme needs and are not discussed further here since our primary focus is on evidence chain generation. After selecting the predictions to be used for therapeutic hypothesis generation, we acquire the associated rules generated by AnyBURL for those predictions. Subsequently, the process of automatic filtering can be initiated.

**Hypothesis generation without filtering.** Once a subset of predictions and their corresponding rules are established from the previous step, we generate evidence chains as shown in Fig. 1b, by querying the graph for all possible paths following the rules that suggested the prediction. Here we show how paths could be generated if no filters were applied. We would only search paths with the types of nodes mentioned in the rule. In Fig. 1b, black box, we call this, "evidence chains without any filters applied" to the AnyBURL output.

**Hypothesis generation with filtering.** Here the auto-filtering model is applied to the subset of predictions and their corresponding rules as shown in Fig. 1d. The Rule-based filter is first applied. As previously described, uninformative rules do not provide a useful biologically relevant therapeutic rationale, and this automatically eliminates drug predictions that were suggested by less relevant rules. Only the resulting predictions and rules go through the phases of significant path filter, deductive path building and the gene/pathway filters as shown in Fig. 1d. More details on filters are provided in Methods. Gene and pathways for filtering are gathered from the biological relevance gathering phase as described below.

**Biological relevance gathering.** We perform an extensive landscape study for the given disease and identify gene-disease associations (GDAs) and biological pathways specific to the diseases of interest as

**Table 1 | Genes and pathways found in evidence chains that have an effect in preclinical experiments in the *Fmr1 KO* mouse model for Ibudilast**

| Gene | P value | Pathway associated | FDR-adjusted pathway P value |
|------|---------|--------------------|------------------------------|
| PDE4D | 0.0276 ↑ | cAMP signalling pathway | 1.02 E-4 ↑ |
| PENK | 0.0434 ↓ | Neuroactive ligand–receptor interaction | 0.0123 ↓ |
| FOS | 0.0602 ↑(significant logFC = 0.854) | cAMP signalling pathway | 1.02 E-4 ↑ |
| AKT1 | 0.029 ↓ | cAMP signalling pathway | 1.02 E-4 ↑ |
| MEF2C | 0.0287 ↑ | cGMP-PKG signalling pathway | 0.0184 ↑ |
| NPY2R | 0.0094 ↓ | Neuroactive ligand–receptor interaction | 0.0123 ↓ |

Columns are *Gene*; HGNC gene symbol, *P value*; statistical significance of the gene being differentially expressed based on an estimate of log fold change following drug treatment. Values computed by DESeq2, *P* values are two-sided. *Pathway associated*; pathway term describing gene function following analysis by gene set enrichment analysis, *FDR-adjusted pathway P value*; statistical significance of pathway enrichment score corrected for multiple hypothesis testing using Benjamini–Hochberg procedure.

**Table 2 | Genes and pathways found in evidence chains that have an effect in preclinical experiments in the *Fmr1 KO* mouse model for Sulindac**

| Gene | P value | Pathway associated | FDR-adjusted pathway P value |
|------|---------|--------------------|------------------------------|
| PTGS1 | 0.0269 ↓ | Serotonergic synapse | Not significant |
| MAPK3 | 0.0028 ↓ | Pathways of neurodegeneration | 4.55 E-24 ↓(Most significant "Alzheimer's disease") |
| PTGS2 | 0.0042 ↓ | Alzheimer disease | 4.55 E-24 ↓ |

Columns are *Gene*; HGNC gene symbol, *P value*; statistical significance of the gene being differentially expressed based on an estimate of log fold change following drug treatment. Values computed by DESeq2, *P* values are two-sided. *Pathway associated*; pathway term describing gene function following analysis by gene set enrichment analysis, *FDR-adjusted pathway P value*; statistical significance of pathway enrichment score corrected for multiple hypothesis testing using Benjamini–Hochberg procedure.

shown in Fig. 1c for cystic fibrosis, Parkinson's disease and FXS. GDAs and pathways are sourced from Genomics England PanelApp[32], Open Targets[33], Pharos[34], Geneshot[35] and Healx KG. GDAs that were found in at least four of the five resources are selected for manual validation. Manual curation assessed the validity of the associated evidence provided by the original resource, and further scientific literature searches were performed when this evidence was insufficient. In order to obtain a list of disease-relevant pathways, we used GDAs curated in the previous step to perform pathway enrichment analysis using Fisher's Exact Test method to map sets of genes to pathway terms[36]. Furthermore, we have applied Benjamini–Hochberg correction[37] to account for multiple hypotheses testing and then used an adjusted *P* value threshold of 0.01 to extract the most statistically significant pathways. Pathways were sourced from KEGG[38], Reactome[39] and Wikipathways[40] and are defined as a series of interconnected biochemical reactions that occur within a cell or organism, leading to a specific biological outcome. As described in Fig. 1c, the resulting genes and pathways are fed into the gene and pathway filter in the pipeline, providing biological context for the automated evidence chains filtering process. Paths that contain at least one of the genes or pathways are retained.

**Rule-based filtering in practice.** Finally, as described in Fig. 1d, we arrive at a reduced number of evidence chains after the application of all filters. To summarize, we compute the reduction achieved via autofiltering, by computing all possible 2-hop paths between the predictions and the disease, paths from all rules produced by AnyBURL for the predictions and deducting the final paths produced by the pipeline. We present and discuss the percentage of reduction achieved for cystic fibrosis and Parkinson's disease case studies in "Results" further below in Table 3.

**Fragile X syndrome case study**
We validated the utility of our automatic filtering pipeline against FXS preclinical experimental data. Briefly, Sulindac and Ibudilast emerged as promising compounds in several non-Knowledge Graph (KG)-based drug prediction models, ranking highly, and with high confidence as potential treatments for FXS. Furthermore, these drugs were found to improve behaviour and cognition in FXS mouse models (*Fmr1* KO1 and *Fmr1* KO2) during in-house efficacy studies. However, the non-KG predictive models lacked the ability to provide a therapeutic rationale for their predictions. In this case study we show how our unique pipeline excels at generating biologically useful evidence for these drug predictions, even when they originated from different models. The only requirement is that query drugs must also be predicted by the AnyBURL model.

We compared the evidence chains extracted for FXS against available preclinical experimental data to confirm that the pipeline automatically extracted mechanistically significant and meaningful information with the potential to inform preclinical decision-making. We analysed the gene expression levels and pathways inferred from FXS mouse models treated with Sulindac and Ibudilast and checked whether gene expression levels were changing in line with expectations of the evidence chains. The preclinical experimental details are provided in "Methods". Results are presented in Tables 1 and 2, which detail the significantly regulated genes and associated pathways for each treatment. For Ibudilast, there is 622 times upregulation and 713 times downregulation in *P* values. In Sulidac, its 1637 times upregulation and 1578 times downregulation. Here, we are demonstrating the observed molecular effects of predicted reasoning from evidence chains observed for FXS and not the entire list of genes altered in the preclinical testing. It is an experimental confirmation of the in silico hypothesis of the mechanism. In addition, Fig. 3a, b presents the automatically generated evidence chains, showing the pathways and significant genes for each drug which were reviewed by experts and considered significant.

In Fig. 3a, we observe that the cAMP (cyclic adenosine monophosphate) signalling and cGMP (cyclic guanosine monophosphate)-protein kinase G (PKG) signalling pathways have been extracted by our pipeline in the paths connecting Ibudilast to FXS. The phosphodiesterase (*PDE*) inhibitor Ibudilast has been shown to inhibit *PDE3A* (cAMP), *PDE10* (cAMP & cGMP), *PDE11* (cGMP), and *PDE4* (cAMP) with preferential potency against *PDE4*[41] and has shown to have several beneficial effects in the brain[42–44]. FXS patients have reduced cAMP levels[45] and several preclinical and clinical studies have supported

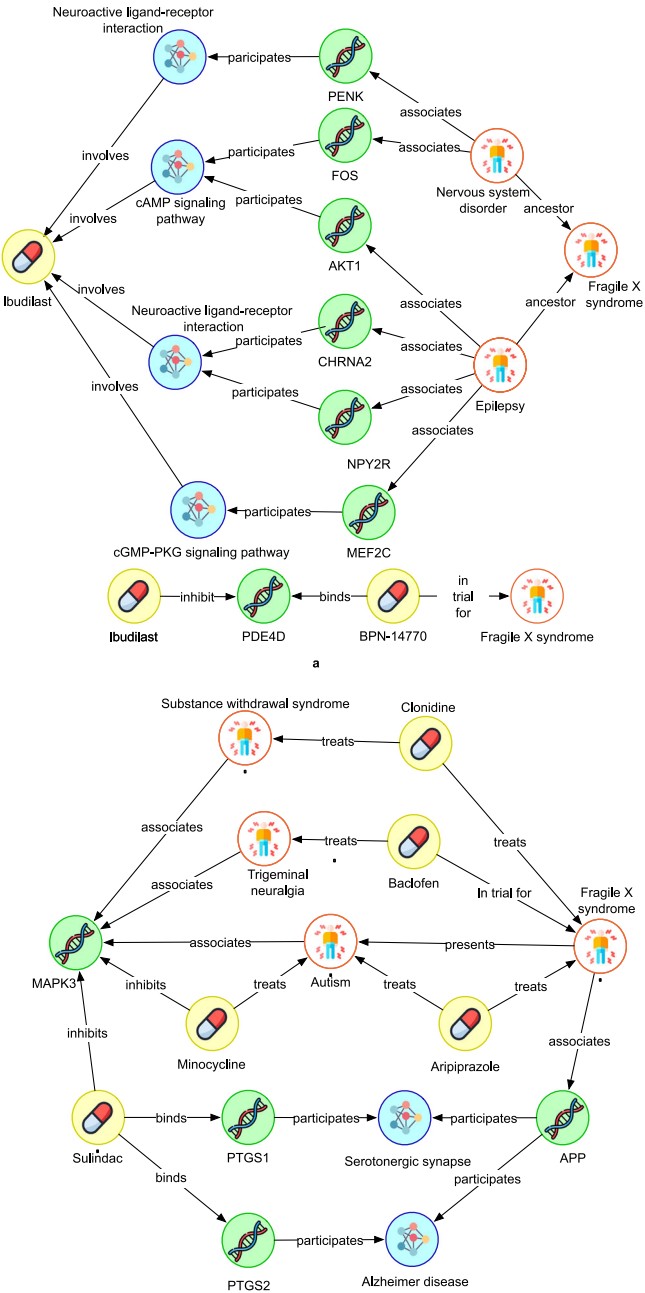

**Fig. 3 | Evidence chains produced for Ibudilast and Sulindac and validated against preclinical experimental data. a** Evidence chains extracted for Ibudilast by the evidence generation and auto-filtering pipeline. **b** Evidence chains extracted for Sulindac by the evidence generation and auto-filtering pipeline. Yellow symbol indicates the type compound, white indicates disease, green indicates gene, red indicates phenotype and blue indicates pathway.

*PDE4* inhibition as a viable target in FXS[46–49]. Inhibition of *PDE10*, levels which are elevated in FXS[50], has also demonstrated efficacy in a preclinical model of FXS by normalising EEG brain activity[51]. The selectivity of ibudilast against both *PDE4* and *PDE10* make it an attractive therapeutic candidate for FXS.

BPN-14770, a specific *PDE4D* inhibitor, improved behaviour and cognition in a mouse model of FXS[52]. In addition to this, BPN-14770 has recently been shown to improve cognition in FXS patients in a phase 2 trial[49]. Ibudilast displayed a similar improvement in cognition and behaviour in an FXS mouse model, likely due to the significant

elevation in the cAMP signalling pathway ($P < 0.0001$) reported in Table 1. Evidence chains also predict Ibudilast as a treatment via *c-FOS* gene and a non-significant elevation in *c-FOS* was also observed following Ibudilast treatment in the disease models as shown in Table 1 ($P = 0.062$). *c-FOS* gene is an early response gene that acts as a transcription factor and is upregulated in response to cAMP-dependent CREB activation following an increase in neuronal activity.

Upregulation of *c-FOS* leads to an elevation in the expression of downstream proteins necessary to strengthen neuronal connectivity to allow for memory formation. *c-FOS* levels are reduced in FXS mouse models as a result of reduced cAMP signalling[53]. In contrast, our gene expression data demonstrated that *PDE4D* expression was significantly elevated ($P = 0.0276$) in the cortex following 3 mg/kg Ibudilast treatment (Table 1). This could simply be a compensatory mechanism, whereby chronic inhibition of the enzyme results in an elevation in its expression. This has been reported for rolipram, another *PDE4* inhibitor, whereby rolipram treatment significantly elevated cAMP levels while paradoxically significantly elevating *PDE4* levels[54].

The evidence chains also predict Ibudilast as a possible treatment for FXS through proenkephalin (*PENK*). *PENK* is an endogenous opioid polypeptide hormone that has been implicated in neuroinflammation and identified as an early indicator of vascular dementia[55]. Although neuroinflammation in FXS is still a contentious topic, microglia and astrocytes from FXS mice have been shown to produce an elevated proinflammatory cytokine response when activated[56,57]. We found that Ibudilast treatment significantly reduced *PENK* expression, suggesting a reduction in neuroinflammatory pathways. Ibudilast has been reported to have anti-inflammatory properties through cAMP signalling, *TLR4* inhibition, and has been shown to protect against reactive oxygen species, a common precursor to inflammation[58–60].

From the evidence chains we see that Ibudilast, through its association with modulating both the cAMP and cGMP pathways, is linked to FXS through the proteins *AKT1* and *MEF2C*, and their association with epilepsy (Fig. 3a). Although not all FXS patients develop seizures, preclinical and clinical studies have demonstrated that by targeting one particular pathophysiological pathway, such as seizures, can often alleviate multiple symptoms in FXS[61–66]. It therefore stands to reason that a small molecule which is able to alleviate seizure incidence could improve other symptoms in FXS and therefore warrants further investigation.

From literature it is evident that inhibiting *AKT1* prevents epilepsy in a rat model of temporal lobe epilepsy[67], it therefore seems rational that modulating AKT signalling through Ibudilast treatment could reduce seizures in FXS patients. We see that 3 mg/kg Ibudilast treatment significantly reduced *AKT1* expression ($P = 0.029$) in the cortex of *Fmr1* KO2 mice (Table 1). *MEF2C* haploinsufficiency has been linked to an increase in seizure frequency and epilepsy[68]. From our internal data we found that cortical *MEF2C* expression was significantly elevated ($P = 0.0287$) in *Fmr1* KO2 mice treated with 3 mg/kg Ibudilast (Table 1). This data suggests that elevated *MEF2C* expression following Ibudilast treatment could reduce seizure frequency in FXS.

The evidence chains also indicate a connection between Ibudilast and FXS through epilepsy, mediated by the neuropeptide Y receptor protein *NPY2R* (as shown in Table 1). *NPY2R* gene, which regulates anxiety, sleep, appetite, and neuronal excitability, has been found to have elevated expression levels in brain biopsies from patients with temporal lobe epilepsy[69]. Administration of 6 mg/kg of Ibudilast significantly decreased the cortical expression of *NPY2R* ($P = 0.0094$) in *Fmr1* KO2 mice, as indicated in Table 1.

Diving deeper into to the evidence chains, Sulindac, an inhibitor of *PTGS1* and *PTGS2*, has been predicted as a potential therapy for FXS by linking these two proteins to their involvement in Alzheimer's disease and amyloid precursor protein (APP) processing (Fig. 3b). APP is a precursor to the toxic amyloid beta protein which eventually forms the amyloid plaques associated with Alzheimer's disease. The expression

of *PTGS1* and *PTGS2* changes throughout the progression of Alzheimer's disease pathology and is believed to contribute to the neuroinflammatory aspect of the disease[70]. To this end, *PTGS2* inhibition has demonstrated therapeutic benefit in preclinical models of Alzheimer's disease. FXS patients also have elevated amyloid beta levels as a result of an increased expression of APP. Modulating *PTGS1/2* levels with Sulindac could therefore be a beneficial treatment for FXS. As conformation of this, Sulindac was found to normalize behaviour and improve cognition in a mouse model of FXS and significantly reduce the expression of both *PTGS1* ($P = 0.0269$) and *PTGS2* ($P = 0.0042$) in the cortex of *Fmr1* KO1 mice (Table 2).

Evidence chain analysis also shows that Sulindac can potentially affect the *MAPK3* signalling pathway, which has been implicated in the symptoms and pathophysiology of FXS. Minocycline in particular, has shown to effectively improve social and cognitive deficits, which are thought to be driven by improved spine maturation, in preclinical models of FXS[71,72]. In Fig. 3b, minocycline and Sulindac have shared profile of both inhibiting *ERK1* (*MAPK3*) activity and its subsequent downstream signalling cascade, further supported by previously published works[73,74].

Clonidine, an alpha2 adrenergic receptor agonist, has been demonstrated to effectively treat symptoms of ADHD in individuals with FXS[75]. It has also been used effectively to manage substance withdrawal[76]. Clonidine works, in part, by modulating downstream signalling via *MAPK3*[77]. In addition, *MAPK3* signalling is also involved in the physiology of substance withdrawal[78]. The link between clonidine and *MAPK3* through substance withdrawal and FXS supports the rationale behind why Sulindac can be a possible treatment for FXS (Fig. 3b).

Baclofen, a *GABAB* receptor agonist, has shown efficacy in FXS mouse models and mixed results in patients[79], where it failed primary outcome measures but met certain secondary endpoints. Reasons for this are thought to be due to small cohorts and lack of patient stratification, rather than poor efficacy of the drug[80]. Baclofen has also been found to be effective in treating trigeminal neuralgia through COX2 inhibition[81,82]. Trigeminal neuralgia has been associated with *MAPK3* signalling[83], which is the same signalling pathway linking Sulindac as a possible treatment for FXS (Fig. 3b). In addition, Aripiprazole, has been effective in treating both FXS and idiopathic autism[84] and has been shown to modulate *MAPK3* signalling[85,86], which is what links this drug to Sulindac.

Overall, our evidence chains imply that Sulindac might be a good treatment for FXS due to its link through *PTGS1*, *PTGS2* and *MAPK3* signalling. These suggestions from our evidence chains analysis are further validated using internal in vivo data, as we observed 5 mg/kg Sulindac significantly reducing the expression of *MAPK3*, *PTGS1* and *PTGS2* in the cortex of *Fmr1* KO1 mice (Table 2). Further to the previous discussion, additional literature evidence suggests that reducing *MAPK3* signalling has benefits in alleviating the symptoms and pathophysiology in preclinical models of FXS[87–90], which further strengthens the use of Sulindac as a potential treatment for FXS.

### Parkinson's disease and cystic fibrosis case study

Typically, drug discovery projects focus on uncovering novel drug-disease relationships and forming hypotheses regarding the potential effectiveness of modulating that link, for example with a small molecule. However, here we intend to establish the performance and validate the utility of our evidence chains filtering approach by rediscovering a set of known treatment relationships along with evidence chains that may explain the therapeutic basis of the treatments. We are presenting these results as a means of validating the method's output as baseline justification, demonstrating its capability to reconstruct information that is already well-known in the scientific literature. To achieve this, we collaborated with curators who performed rigorous manual curation of the literature to compile a comprehensive list of approved treatments for cystic fibrosis and Parkinson's diseases. These treatments were curated based on their relevance to the disease itself or its associated symptoms (phenotypes). In total, we identified 45 approved treatments for Parkinson's disease and 17 for cystic fibrosis. Prior to running experiments, we removed all direct links between the approved treatments and the disease in the knowledge graph. Our pipeline was able to predict these approved treatments (44 out of 45 approved drugs for Parkinson's disease and all 17 treatments for cystic fibrosis) in the absence of known links and successfully generated the corresponding evidence chains for those treatments.

From the extensive list of initial rules generated by AnyBURL for both case study disease predictions, we undertook a manual curation effort involving collaboration with drug discovery scientists, who identified rules with the potential to yield biologically relevant evidence chains, as previously described. Our automated filtering model retained all the curated rules for both diseases. This outcome affirms that the method effectively preserved valuable information, ensuring that it was not overlooked prior to the path generation process.

Figure 4 illustrates evidence chains generated for cystic fibrosis (Fig. 4a) and Parkinson's disease (Fig. 4b) for few of the approved treatments predicted. Cystic fibrosis (CF) is a rare genetic autosomal recessive disorder caused by mutations in the cystic fibrosis transmembrane conductance regulator (*CFTR*) gene. Notably, the importance of *CFTR* for cystic fibrosis is also observable in Fig. 4a, which clearly demonstrates the ability of our evidence chains methodology to identify the most pertinent information about the disease.

Our evidence chains show both Mannitol and Tobramycin as potential treatments for cystic fibrosis, as they have been traditionally used to treat bronchiectasis, which can be caused by chronic infections or cystic fibrosis[91]. QBW251 (Icenticaftor), which functions as a potentiator of the *CFTR* protein by binding and unlocking the *CFTR* channel to facilitate chloride ion transport[92], has also shown clinical efficacy in treating cystic fibrosis and chronic obstructive pulmonary disease[92–94]. Similarly, our predictions identified Ivacaftor, Tezacaftor, and Lumacaftor as potential treatments due to their similar mechanism of action with QBW251. Lumacaftor's link to cystic fibrosis in our evidence chains was through improving pancreatic functioning in acute pancreatitis and this is a symptom associated with the disease[95,96].

Interestingly, we also found that our unique auto-filtering approach can generate potentially valuable rules and evidence that, while not originally biologically significant to the disease of interest, might still be useful for further experimentation in some cases and even uncover novel rationale that is not known. We showcase this with cystic fibrosis. Specifically, Acetylcysteine, an approved cystic fibrosis treatment, was found to have a significant number of evidence chains linking it to the disease. Figure 4a displays examples of strong evidence chains indicating that Acetylcysteine could be beneficial in addressing pulmonary fibrosis symptoms, which is a common complication in cystic fibrosis patients suggesting it may be a potential treatment. However, in Fig. 4c it becomes difficult to determine whether Acetylcysteine is a viable candidate for treating cystic fibrosis as the evidence chains shown in Fig. 4c are more indirect. The first evidence chain in Fig. 4c suggests that both Acetylcysteine and Tenapanor are being tested as treatments for end-stage renal failure. Kidney failure is a complex health issue that can arise due to a variety of factors, so the fact that two compounds are being tested as treatments for this condition does not necessarily mean that they will be effective treatments. Secondly, there are 4367 interventional clinical trials for kidney failure listed in Clinicaltrials.gov, so the fact that a compound is being tested for this condition does not necessarily mean that it is an approved and reliable treatment. Therefore, compared to the evidence presented in Fig. 4a, the evidence presented in Fig. 4c is not as strong or biologically

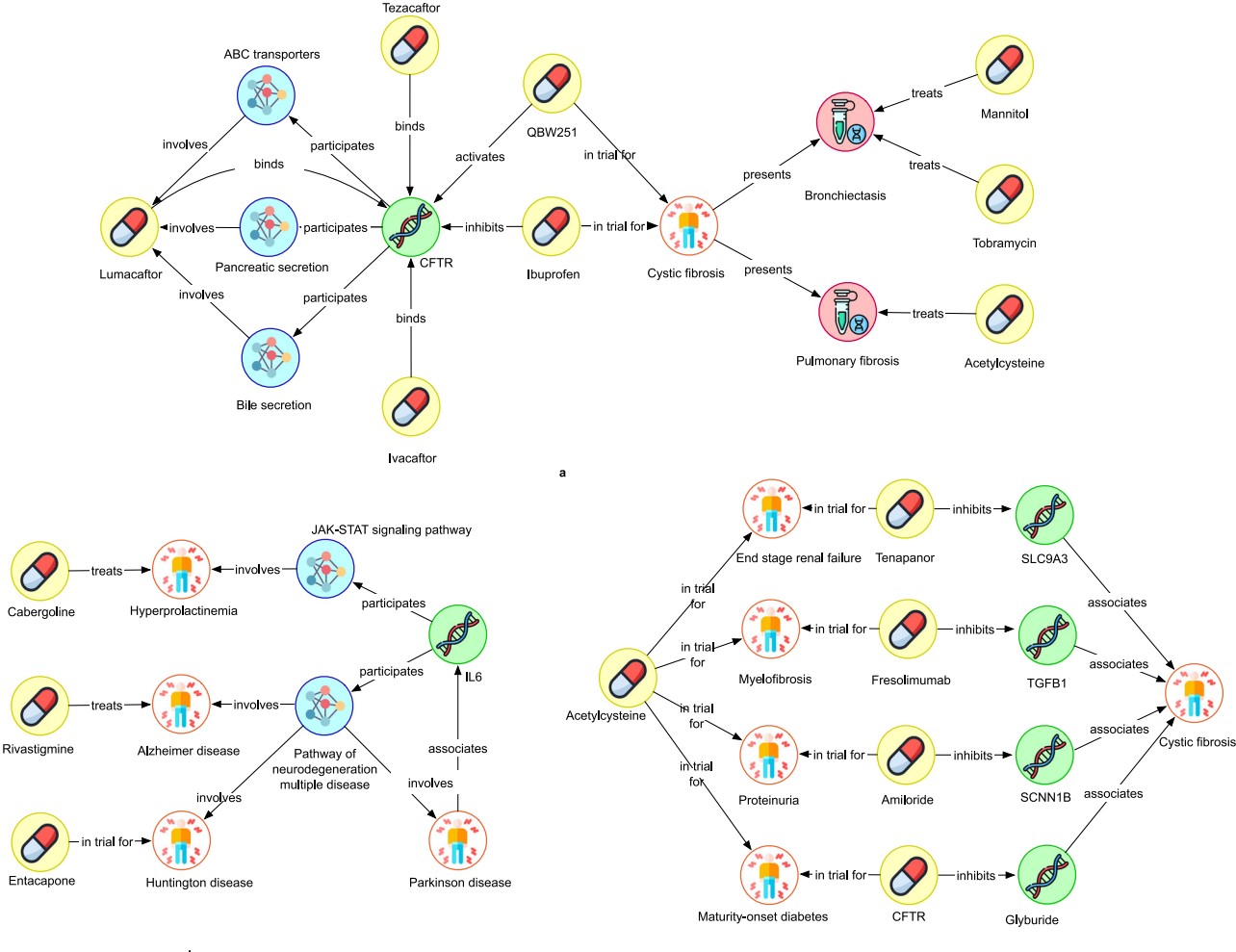

**Fig. 4 | Evidence chains linking approved treatments to the case study diseases.**
**a** Most significant evidence chains extracted for cystic fibrosis **b** Most significant evidence chains extracted for Parkinson's disease **c** Evidence chains linking treatment drugs to cystic fibrosis that are less significant. Yellow symbol indicates the type compound, white indicates disease, green indicates gene, red indicates phenotype and blue indicates pathway.

relevant. However, for compounds and diseases for which there is not any strong direct evidence like the ones presented in Fig. 4a, the less biologically relevant evidence presented in Fig. 4c may still provide useful insight for further experimentation.

Parkinson's disease (PD) is a neurodegenerative disorder characterized by inflammation and oxidative stress, which play critical roles in its pathogenesis. Looking at the predictions and corresponding evidence chains in Fig. 4b, Cabergoline is suggested as a potential treatment for the disease, mainly due to its link with hyperprolactinemia and the JAK/STAT signalling pathway. Cabergoline, with its potent D2 selectivity, is currently an effective treatment for hyperprolactinemia[97]. The pathophysiology of Parkinson's disease is associated with increased neuroinflammation, such as *IL6* signalling through the JAK/STAT pathway.

Rivastigmine, a cholinesterase inhibitor, used for the treatment of Alzheimer's disease was also predicted as a treatment for Parkinson's disease according to Fig. 4b. In the evidence chain analysis, Alzheimer's disease is linked to it through the inflammatory cytokine, *IL6*. Commonly seen in various neurodegenerative diseases, neuroinflammation and inflammatory cytokines are important drivers of pathophysiology, and there is strong evidence that inflammatory cytokines, such as IL6 are involved in Parkinson's disease and Alzheimer's disease progression[98,99]. The evidence chains show Entacapone as a treatment

through a link with Huntington disease, another neurodegenerative condition linked with ataxia.

Although the evidence shown here are for already approved treatments, we have included them as means to validate the approach and to sanity check the extracted evidence chains with already known information. Overall, all these evidence chains and their corresponding supporting information demonstrate the capability of our methodology to capture essential information for identifying potential treatments for diseases and to provide critical insights into how those drugs may treat the diseases.

**Reduction in the number of paths generated**

To assess the overall efficiency of the automated pipeline, we compute all possible paths in the Healx KG up to 2-hop length between the approved treatment drugs and the disease for Parkinson's disease and cystic fibrosis. In total 719,203 paths for Parkinson's disease and 310,901 paths for cystic fibrosis were computed as shown in Table 3. We also compute the total number of paths for each approved drug by using all rules given by AnyBURL for the drug without any filtering applied. Table 3 shows the number of all possible 2-hop paths that could be generated, the number of paths extracted from rules given by AnyBURL, the number of paths after application of the auto-filtering model and the total reduction achieved for all the approved drugs

**Table 3 | Reduction in evidence chains space with auto-filtering pipeline**

|  | 3-hop possible paths | No. of AnyBURL paths | No. of filtered paths | Total reduction |
|---|---|---|---|---|
| Parkinson's | 719,203 | 156,254 (562k reduction) | 34,738 (121k reduction) | 684 K |
| Cystic fibrosis | 310,901 | 112,276 (198k reduction) | 44,025 (68k reduction) | 266 K |

listed for both diseases. In Fig. 1, we showed a complete workflow of how these evidence chains are generated with and without the auto-filtering method. With the automated filtering approach, we achieved a 77% reduction in evidence chains compared to the ones generated by AnyBURL for Parkinson's disease resulting in a total of 95% reduction compared to all the possible paths that can be generated. Similarly, for cystic fibrosis, we achieved a 60% reduction compared to AnyBURL and in total 85% compared to the full list of paths. However, if this were a novel drug discovery programme, then there would still be a prohibitively large number of paths to explore by human experts.

In fact, the case studies presented here already have approved treatments and these nodes are highly connected in the graph. Even though we removed the direct links between the treatments and the disease in the input graph, it still produces a significant number of paths. In disease case studies without an approved treatment, we find that these numbers are lower compared to what is shown in Table 3. For example, in the FXS case study the method extracted 249 evidence chains for Sulindac and 324 for Ibudilast which is feasible to review by experts and demonstrates the applicability of the approach in diseases without an approved treatment.

## Discussion

We have demonstrated the usefulness of our automated methodology for distilling informative evidence chains useful in drug discovery through the study of Fragile X Syndrome, a rare genetic disorder and two other diseases, Parkinson's disease and cystic fibrosis. Our contribution enables KBC models such as AnyBURL to be applied in establishing pharmacological rationale through the reduction of uninformative paths without loss of biological signal. We started with a symbolic model AnyBURL applied to the Healx KG to make treatment predictions and generate useful evidence chains explaining indicated treatments with an automatic filtering approach (explained in "Methods"). Our results from the FXS study demonstrate a strong correlation between the evidence chains and experimentally derived transcriptional changes in essential genes and hallmark pathways of the drug treatments indicating that the drugs may be causally linked with therapeutic benefit in the disease model. In both cystic fibrosis and Parkinson's disease, we found the most relevant genes and pathways extracted automatically by the evidence chains. The approach can be applied consistently across multiple diseases, eliminating the need for manual curation of rules or evidence in the process. This clearly highlights the potential of automating the process of generating therapeutic rationale for drug predictions. We suggest this approach enables the power of tailored reinforcement learning to be applied more frequently as a tool for rapidly deriving mechanistic insights, reducing the time and cost investment required for laboratory experiments, and guiding further clinical development.

As shown in Table 3, we achieved a 77% reduction in the paths generated for Parkinson's disease and 60% for cystic fibrosis with the automatic filtering approach alone. While this still leaves many paths to explore for highly connected diseases in the graph, such as those with approved treatment, for rare disease case studies like FXS we find that these numbers are amenable to human triage. As future work we intend to reduce this space more by using additional rules that make use of confidence scores. Although rule confidence scores are produced by AnyBURL they do not translate to biological relevance. Rules with the highest confidence scores are not always the most significant or are too vague or generic from a drug discovery perspective. It is also important to note that these rules involve many independent degrees of freedom with their own assumptions. High confidence relations between any two given entities $A$ and $B$ and $B$ and $C$ does not imply a high confidence relationship between $A$ and $C$. However, for future work assigning a rule or path confidence score is vital and could be done using better path ranking methodologies or even embedding evidence weights in the training process of the model, so that irrelevant chains are penalised pushing the model towards generating even more biologically relevant rules and paths.

As discussed briefly in the Introduction section there are no benchmark datasets available to evaluate evidence chains generated for a given drug prediction in a specific disease. Creating a gold standard validation set of evidence chains for a list of drug-disease pairs would require decision-making on what is the rationale for choosing the top-ranking evidence chains, differs for each disease of interest, and therefore would require an inordinate amount of time to conduct expert review from curators, pharmacologists and bioinformaticians. As future work we plan to create such a benchmark dataset of highly relevant paths for drug-disease pairs to quantitatively validate the approach.

We intend to also integrate data on protein-protein interactions, proteins and biological functions interactions and interactions between biological functions in the knowledge graph to better explain the mechanism of action of the drug in the disease of interest. Another limitation in this work is the use of publicly available, proprietary, and curated data sources alone for the input knowledge graph. Details of the data used in this study is shown in section Data in "Methods". Enormous amounts of data are available from biological text sources processed by our natural language processing (NLP) pipelines, and they extract biological relations between entities. Yet using it here causes bottlenecks in the evidence chain generation and interpretation process due to the large number of rules/evidence that can be generated from this data. As part of our future work, we look for better ways to incorporate data processed by our NLP pipeline.

## Methods

### Data

The data described in this paper producing all experimental results shown in "Results" were sourced from Healx's proprietary Knowledge graph (Healx KG) which includes internally curated edges for rare diseases, compounds, and other entities. It is comprised of various public and commercial data sources including ChEMBL[100], CTD[101], Drugbank[102], HGNC[103], Human Phenotype Ontology[104], KEGG Pathway[38], NCBI Gene[105], Orphanet[29], PanelApp[33], Open Targets[33], Pharos[34], SIDER[106], Geneshot[35], UniProt[107], NDF-RT[108], OMIM[109], Reactome[39], Wikipathways[40], MONDO[28], MeSH[110], Pharmaprojects and together with internally curated data. The internally curated data added to the knowledge graph comprises around 2000 associations relating to diseases, phenotypes, compounds, and targets. All information was curated based on published sources of information and does not require ethical approval. Table 1 in the supplementary file provides a complete list of publicly available and commercial or licensed sources and the versions used in building the Healx KG. Additional information is extracted from public documents and package inserts. All data is stored as a set of triples consisting of head and tail entities connected via a relation type. The Healx KG consists of 8 node types including Compound, Disease, Gene, Protein, Pathway, Mechanism, ATC and Phenotype and 29 edge types. Supplementary

Table 2 provides information on the types of edges in the graph. In Supplementary Fig. 1, we also provide a meta-graph showing the different node and edge types and how they are connected in the Healx KG.

We share a non-confidential version of the Healx KG, a subgraph of the full KG which contains 129,501 edges and 37,331 nodes. This graph does not contain nodes and edges coming from proprietary data sources and internal curation. To enable the reproducibility of the filtering method, we have repeated all experiments with this subgraph for Parkinson's disease. In Supplementary Fig. 2, we have presented a few interesting evidence chains observed for predictions, Rivastigmine, Entacapone and Cabergoline with their connections to Parkinson's disease using this Healx subgraph. We also share the percentage of reduction achieved in Supplementary Table 5 showing the total possible number of 2-hop paths that could be generated for 34 predicted approved treatments for Parkinson's in this graph and the number of paths extracted for those predictions with AnyBURL and the auto-filtering model. On average we observe 97% reduction in the number of paths for the predictions. The main results of the manuscript are produced using the source code we are sharing with the manuscript and the full Healx KG. This accounts for any minor differences in specific evidence chain composition shown in the Supplementary results for Parkinson's disease, yet this does not limit or distract from the main learning of our manuscript regarding reduction in uninformative paths without loss of biological signal and the general applicability of our workflow in drug discovery.

## Anytime bottom-up rule learning (AnyBURL)

The reinforcement learning-based approach in AnyBURL yields an explanation in terms of the rules that make a prediction. The learning process is conducted in a sequence of time spans of length $t_s$. Within a time span the algorithm learns as many rules as possible by iteratively sampling random paths with the goal of covering and satisfying as many training facts as possible within a given time span. A rule takes the form, $h(c_0, c_1) <- b_1(c_1, c_2),\ldots, b_n(c_n, c_{n+1})$ with a ground path rule of length $n$ where the head of the rule is $h(\ldots)$ and $b_1(\ldots)$ through $b_n(\ldots)$ is its body and $c_0$ to $c_n$ are variables that can represent entities in the knowledge graph. Each rule has a confidence score usually defined as the number of body groundings, divided by the number of those body groundings that make the head true. Variables in rules are instantiated with specific entities to obtain rule groundings[111]. This is usually done by instantiating variables in the rule body by filling in the entities whose relational triples match the rule body. A corresponding new triple is obtained by instantiating the rule head. The instantiated rule is called a grounding. The learnt rules are then applied to make predictions given the fact that the predicted entity is supported by at least one rule. This generates a set of predicted candidates for a given query along with the set of rules that suggested it.

Candidate predictions are ordered via the maximum confidence of all rules that have generated the candidates. If the maximum score of several candidates is the same, the candidates are ordered via the second-best rule that generates them, and so on, until a rule is found that makes a difference. AnyBURL can learn rules in a short time, defined usually in seconds as a parameter $t_s$ which is quite competitive compared to many other approaches since unlike embedding models it does not require time-consuming hyperparameter tuning to achieve good performance. The learning and prediction yield a set of ranked candidate predictions and associated rules. In Fig. 1, we have shown examples of rules learnt by AnyBURL. The body of the rule contains relationships in the graph between entities represented by variables forming paths. Each rule shown has a confidence score indicative of the significance to the candidate prediction. However, following review, we find these scores do not translate to a simple measure of biological importance. We note, the highest-ranking rules are not always the most significant, or are too vague or generic from a drug discovery perspective.

$$\text{Compound} - treats \rightarrow \text{Disease} \leftarrow descendant - \text{Disease} \quad (1)$$

$$\text{Compound} - treats \rightarrow \text{Disease} - ancestor \rightarrow \text{Disease} \quad (2)$$

$$\text{Compound} - binds \rightarrow \text{Gene} - participates \rightarrow \text{pathway} - involves \rightarrow \text{Disease} \quad (3)$$

$$\text{Compound} - involves \rightarrow \text{Pathway} - involves \rightarrow \text{Disease} - ancestor \rightarrow \text{Disease} \quad (4)$$

$$\text{Compound} - treats \rightarrow \text{Disease} - associates \rightarrow \text{Gene} \\ \leftarrow associates - \text{Disease} - ancestor \rightarrow \text{Disease} \quad (5)$$

For example, rules (1) and (2) shown above frequently receive high confidence scores compared to other rules but are less significant in the drug discovery context since they only have ancestor and descendant relationships that do not inform any mechanistic understanding of how the drug treats a disease. Although rules (3), (4) and (5) scored low they contain more biologically informative relationships such as disease–gene, gene-pathway associations and so generate more compelling evidence with greater interpretability in a drug discovery context. Therefore, we considered all rules associated with a single prediction to generate evidence for the predictions, not applying any confidence score filtering in terms of rule composition. In all experiments, known treatment relationships between the disease of interest and the drug were removed from the Knowledge graph before training the AnyBURL model on them.

## Evidence generation with automatic filtering

Our proposed methodology utilizes automatic filtering to extract biologically meaningful evidence from a list of predictions and their corresponding rules, with a focus on identifying paths in the graph that are most relevant to the disease of interest which we call the evidence chains. The complete workflow is shown in Fig. 2. The method is flexible and allows for the disabling of certain filters as needed to meet the requirements of the project, making the approach flexible and consistently applied across multiple disease projects. A path, in this context, refers to a sequence of nodes and relations that starts at a compound entity and ends at a disease entity, providing evidence that the compound can potentially be useful in treating the disease. There are four stages to the automated filtering process starting with rule filtering, significant path filtering, deductive reasoning-based path building and gene/pathway filtering as shown in Fig. 2. Each stage is described below.

**Rules-based filtering.** From the initial set of rules suggested for candidate predictions, we filter out rules containing less relevant information in the biological context. Rules containing more than one "*disease_disease_ancestor*" or "*disease_disease_descendant*" relationship will be ignored. Rules containing more than two "*in trial for*", "*in vivo preclinical trial for*" and "*disease _compound has orphan designation for*" relationships will be ignored. Examples of such rules are shown below.

$$\text{Compound} - treats \rightarrow \text{Phenotype} \leftarrow presents - \text{Disease} \\ \leftarrow ancestor - \text{Disease} - ancestor \rightarrow \text{Disease}$$

$$\text{Compound} - in\,trial\,for \rightarrow \text{Disease} \leftarrow in\,trial\,for - \text{Compound} - in\,trial\,for \rightarrow \text{Disease}$$

The underlying rationale for this filter is firstly an evidence chain with more than one ancestor or descendant relationship becomes uninformative when trying to understand a potential treatment like

**Fig. 5 | Significant path-based filtering in evidence chains.** The paths between Levofloxacin and cystic fibrosis showing a *treats* and an *in trial for* link between the drug and the disease Bronchitis. The yellow symbol indicates the type of compound, white indicates disease.

explained in Fig. 1c. Depending on how the graph was constructed and how disease sub-types are categorised in the graph this filter could be altered. In our case with Healx KG, we expected one link of the sub-type with the actual rare disease, so the essential information is still captured in the evidence chains. Secondly, there are typically a wide range of trials, orphan designations or in vivo preclinical trials associated with a given disease, many of which fail at various stages and do not ultimately lead to a treatment for the disease. In contrast to the "compound *treats* disease" relationship, which is restricted to the "approved" drug space, other compound-disease relationships such as *in trial for* or in vivo *preclinical trial for* may be considered more "relaxed". Including more than one "relaxed" edge in the evidence chains may lead to broader and noisier explanations being present in the evidence set. Therefore, paths for each prediction are generated only for the remaining set of rules.

**Significant path filtering.** This filter focuses on retaining only the significant paths. For example, in the case of Levofloxacin treating cystic fibrosis, two paths may be generated as shown in Fig. 5, one with an *in trial for* relationship between the disease Bronchitis and the compound, and another with a *treats* relationship between the same entities. However, the *treats* relationship is considered more significant than the *in trial for* relationship, and only the former is retained during the filtering process. This same approach is applied for in vivo *preclinical trial for* and *has orphan designation for* relationships too. They are ignored if a *treats* relation exists between the same entities in the evidence chain.

**Deductive reasoning-based path building.** In this stage paths which were not already extracted by the pipeline using AnyBURL are deduced using significant paths obtained in the previous stage. These paths still exist in the graph but are not directly extracted by the pipeline, most likely due to not learning a specific rule during the training or the rule not being applied during the prediction stage of the AnyBURL model. In the Levofloxacin example below there are four paths generated by the pipeline as shown. Each path corresponds to a different rule, between the entities Levofloxacin and cystic fibrosis.

Levofloxacin−*in trial for* → Bronchitis ← *treats*−Dornase alfa−*in trial for* → Cystic fibrosis

Levofloxacin−*in trial for* → Bronchitis ← *treats*−Dornase alfa−*treats* → Cystic fibrosis

Levofloxacin−*treats* → Bronchitis ← *treats*−Dornase alfa−*in trial for* → Cystic fibrosis

Levofloxacin−*treats* → Bronchitis ← *in trial for*−Dornase alfa−*treats* → Cystic fibrosis

A graphical representation of the above paths is shown in Fig. 6a. Given that *treats* relationship is more specific than *in trial for* and and it exists between all entities but in different paths, we can deduce the path highlighted in red in the figure. The deduced path is more specific

compared to all five paths extracted as shown in Fig. 6b. This path builder helps to further check the output from previous filters and remove other redundant paths that still exist in the generated paths and to deduce paths that have not been directly extracted by the pipeline.

**Gene- and pathway-based filtering.** Once the paths are generated, we filter for paths that contain at least one of the genes or pathways that are considered mechanistically important to the disease given by bioinformaticians and curators after an extensive disease landscape study for the given disease. How we obtain these Gene-Disease associations is explained under Fig. 2 in "Biological relevance gathering". The rationale for including a gene and pathways-based filter is to reduce the set of evidence chains to those that are more disease-informative and thereby the most likely candidates for targeting with a suitable therapeutic hypothesis. The result is to focus on key genes and pathways, over the list of all possible genes and pathways that can link a drug to the disease. The pipeline is flexible so individual filters could be turned off. If a more comprehensive set of paths are required, the gene/pathway filter could be disabled.

**Preclinical experimental details**
Following are the details of the preclinical experimentation on mice dosing and RNA-seq analysis in FXS case studies. All work in the study involving animals was carried out by a third-party CRO in line with the requirements of the United Kingdom Animals (Scientific Procedures) Act, 1986. Due to Fragile X Syndrome being an X-linked inherited disorder, with the most severe phenotype persisting in males, we only tested male mice. In vivo proof-of-concept studies with Ibudilast (Merck, I0157) and Sulindac (Merck, S8139) confirmed that the drug was efficacious in both the *Fmr1* KO1 and KO2 mouse models. The *Fmr1* KO1 mouse model was developed by insertional inactivation of exon 5 of the *Fmr1* gene, leading to the loss of FMRP expression as described at the Dutch–Belgian Fragile X Consortium[112]. Despite the loss of exon 5, the *Fmr1* gene still has an intact promoter, resulting in an abnormal residual *Fmr1* RNA transcript being expressed. The new *Fmr1* KO2 mutant was developed as an alternate model in which the *Fmr1* RNA transcript expression is abolished, thereby preventing FMRP expression. Both mouse models were extensively validated by others, and to date, no major difference has been reported[113–118]. Healx used the *Fmr1* KO1 mouse model in earlier studies, but it was later superseded by the *Fmr1* KO2 model, which lacks expression of FMRP and any *Fmr1* transcript[119,120].

*Fmr1* KO2 model was inbred at GeN DDI, *Fmr1* KO1 model was inbred at Jackson laboratories. *Fmr1* KO2 mice were backcrossed for at least eight generations to C57BL/6J, and WT C57BL/6 J littermates were used as controls. *Fmr1* KO1 mice were backcrossed for at least eight generations to FVB/n, and WT FVB/n littermates were used as controls. Heterozygous breeding pairs were used to generate WT and KO littermates for all studies. Ten male mice (aged 2 months) were used for each treatment group across all behavioural experiments. Only male mice were used for all experiments because FXS is an X-linked inherited disorder with the most severe phenotype persisting in males. *Fmr1* KO1, *Fmr1* KO2 and WT littermate mice were injected intraperitoneal (i.p.) with vehicle [10% DMSO (Merck, 276855) in 90% (20% Captisol

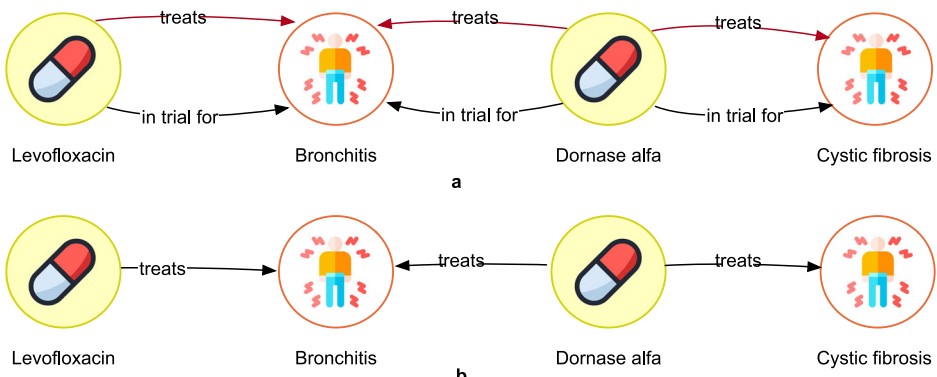

**Fig. 6 | Deductive reasoning applied to evidence chains output. a** Paths existing between the entities before deducing significant paths **b** The most significant path deduced after application of the deductive reasoning-based path builder. Yellow symbol indicates the type compound, white indicates disease.

(Selleckchem, S4592) in Saline)] or ibudilast or sulindac for 2 weeks. All drugs were formulated and dosed in the vehicle solution. Administration volumes were 3.85 mL/kg, such that an adult mouse weighing 26 g received a 0.1 mL injection volume. For all test articles, the volume to be administered was based on each mouse's body weight. Animals were housed in groups of 4 animals per cage, of the same genotype in a temperature- and humidity-controlled room with a 12-h light–dark cycle (lights on 7 a.m.–7 p.m.). Mice were housed in commercial plastic cages (40 × 23 × 12 cm) with Aspen bedding and without environmental enrichment on a ventilated rack system. Food and water were available ad libitum. Experiments were conducted in line with the requirements of the United Kingdom Animals (Scientific Procedures) Act, 1986. To identify transcriptomic changes as a result of drug treatment, *Fmr1* KO2 mice were dosed with Sulindac (2.5 mg/kg or 5 mg/kg) and *Fmr1* KO1 mice were dosed with Ibudilast (3 mg/kg or 6 mg/kg) for 2 weeks. Following dosing mice were cervically dislocated before dissecting out the hippocampus and cortex for storage at −80 °C. Brains were transported to Eurofins genomics for RNA extraction and RNA-seq generation. RNA-seq reads were aligned to the Mouse B38 genome (mm10) using OmicsoftGenCode V24. Low read count genes were filtered out (1 CPM in at least one sample, median <10 across all samples). Gene count matrices were loaded into R (V4.2.0). Differential gene expression analysis was performed using DESeq2[121]. Gene set enrichment analysis (GSEA) was performed using a ranked gene list from the DESeq2 results using fgsea[122]. Genes were ranked by logFC and raw *P* value. Statistically significant thresholds were abs(logFC) >0.58 and *P* value <0.05. Pathway information was taken from KEGG.

### Reporting summary
Further information on research design is available in the Nature Portfolio Reporting Summary linked to this article.

## Data availability
In Supplementary Table 1, we provide details of both public and commercial data sources used in Healx KG. CTD[101], SIDER[106], DrugBank[102], KEGG[38], OMIM[109] and Pharmaprojects are commercial data sources. CTD[101] data can be used only for research and educational purposes, and any Commercial users are required to purchase a license to access data from the CTD website. SIDER[106] data is licenced under a creative commons Attribution-Noncommercial-Share Alike 4.0 License. For commercial use or customized versions, license should be obtained from biobyte solutions GmbH. Use and re-distribution of the content of DrugBank[102] for any purpose requires a license. Academic users may apply for a free license for certain use cases and all other users require a paid license. KEGG[38] database is available for academic use but any commercial use requires a license. Use of OMIM[109] is provided free of charge to any individual for personal use, for

educational or scholarly use, or for research purposes through the front end of the database. Commercial users who want to download all or part of OMIM must obtain a license by paying applicable licensing fees to and entering into a license agreement with Johns Hopkins University (JHU). Pharmaprojects comes with a commercial license granting full access to their APIs. We have shared a subgraph of the Healx KG data created for reproducibility purposes and is available in github, https://github.com/healx/automated-biological-evidence-generation-in-drug-discovery[123]. We have shared the experimental results from this subgraph showing a few interesting evidence chains generated for Parkinson's disease in Supplementary Fig. 2 and the percentage of reduction achieved in Supplementary Table 5. The source data for this is provided as a Source Data file. The raw sequence (RNA-seq) data used in the Fragile X study has been deposited in the NCBI Sequence Read Archive (SRA)) under BioProject PRJNA1096445 titled 'Brain-specific gene expression changes in FXS mouse model after Sulindac or Ibudilast treatment'. The list of curated rules from the complete set given by AnyBURL as given by drug discovery scientists for both Parkinson's disease and cystic fibrosis are included in Supplementary Table 3. The rules set produced for predictions in both diseases have overlaps, hence they are presented in a single table. We also present FXS rules separately in Supplementary Table 4 which were automatically generated by the pipeline before evidence chain generation. The curated list was only used in Parkinson's and cystic fibrosis to check if the automatic filtering retained these useful rules. All rules shared in the supplementary files are from the full Healx KG. Source data are provided with this paper.

## Code availability
The Python implementation of our methodology is available at https://github.com/healx/automated-biological-evidence-generation-in-drug-discovery[123]. The AnyBURL v21 Java model was trained using OpenJDK v14 and small bug fixes to this model have been made and are available in the Github repository. All downstream analyses were performed using Python 3.11, with additional packages used are 'attrs', 'click' and 'numpy'. Please read the README document for information on downloading and running the code.

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

## Author contributions

S.S. designed and performed research including analysis of the data with the method; B.O. performed critical review and interpretation of the results, designed the automatic filtering pipeline, curated ground truth rules for disease case studies; S.S. and J.C. contributed to the creation of

the software pipeline including evidence chains generation and automatic filtering; D.O'D contributed to the subgraph creation and the code repository enabling reproducibility of the results; W.C., A.C. and E.T. analysed evidence chains and preclinical experimental data for FXS and interpreted results; S.S., B.O., W.C. and I.R. wrote the paper; N.T. and I.R. reviewed all experimental results for publication.

## Competing interests

The authors declare no competing interests.
