## [Peer Review File · Nature Communications]

Reviewers' Comments:

Reviewer #1:

Remarks to the Author:

Sudhahar, Ozer, Chang et al present a topical and interesting research article exploring the value of biological knowledge graphs in predicting and explaining repositioning hypotheses in key disease segments. The authors describe a graph completion model to identify the most relevant 'rules' that could explain the biological rationale and mechanism-of-action linking a therapeutic intervention to disease. The use-cases discussed are in Parkinson's Disease, Cystic Fibrosis and Fragile X Syndrome, and I commend the team for delivering in vivo validation studies for Fragile X that are rare to find.

Whilst the study is well written and discusses a critical unmet need on available therapeutic interventions in Fragile X syndrome, I found the overall structure of the manuscript confusing and at times hard to follow the narrative and fully appreciate its novelty/impact. I believe the manuscript needs an overall rethink. Given the therapeutic space of rare genetic disorder this study covers, I do believe this manuscript could be very beneficial to the community and rare disease patient groups in general. With this in mind, I recommend the authors revisit the messaging and flow of the manuscript taking these major concerns/feedback into consideration.

Major:

- My primary suggestion would be for the authors to change the focus of the manuscript to Fragile X Syndrome - discussing the predictive analytics, data that was fed into the KG that resulted in these insights, critical data gaps that exist in the rare disease world, the mechanistic explanation for predictions, followed by in vivo validation. Most of these are already in the manuscript currently, but seems like an after-thought rather than the punchline which I believe it should be.
- The results presented for Parkinson's Disease and Cystic Fibrosis are neither novel nor mechanistically very interesting from a Knowledge Graph-driven insights perspective. These are just observations from prior knowledge that don't need predictive power. While I understand that this was presented as a 'baseline' in the absence of a gold standard, I don't think it is required to be presented as a key result which I do not think it is.
- Methodological innovation came across as very weak, with reimplementing of established methods like AnyBURL without any detail of what is the advancement rather than reimplementing this study delivers. The automated rule filtering method was called out as 'novel' by the authors, which seemed more logical/hierarchical than statistical. I might've missed the key point here, but if the authors wanted to project this as a novel aspect of this study, it should be a key part of the Results with a workflow diagram readers can learn from and reimplement, and not confined to Methods.
- Lastly, as a reader, other than the insights discussed in the paper, there is no other usable outcome that I could take away from the paper. I would like to see a compilation of data, a snapshot of the KG that delivered these results (minus the IP restricted parts), a function/library for the automatic rule filtering approach, and/or a dictionary of rules that rare disease scientists and patient cohorts can learn from.

Minor:

- The manuscript needs a deeper background as to why KGs are relevant in this space, and what has already been achieved. With the recent ongoing focus/scrutiny of AI, it is key to discuss where the authors believe value lies and what the key gaps are as identified by previous studies in this space from a technical (KG) as well as a scientific (repositioning in rare diseases) perspective.
- References are missing for key statements stated as facts
- The definition of 'significance' is unclear across the manuscript, from the figures it seems logical rather than statistical driven by a scoring function. I might be right or wrong, but this needs clarification with better phrasing.
- For the data sources feeding into HealxKG, important for the readers to see a dictionary of how much and what data each of these are contributing, along with last version date. At least one data source is very out of date and needs updating.

I have also included a document with detailed comments and suggestions for the authors to consider and address, wherever appropriate.

Reviewer #2:

Remarks to the Author:

This paper "An experimental validated approach to automated biological evidence generation in drug discovery using knowledge gaps" by Sudhahar and colleagues from Healx describes a methodology to prioritize drug discovery for rare diseases. The rationale is that for many rare and genetic disorders, much information is available scattered around databases such as clinical symptoms, affected pathways, animal models potential therapeutics. By linking these databases using a variety of AI-based computational knowledge, novel drug candidates can surface. Typically, an overwhelming number of potential drug targets is predicted that require time-consuming manual curation by experts in the disease. The authors here present a methodology to automatically reduce the number of pathways predicted and so make drug predictions more accurate and less laborious.

Needless to say such a system is of added value to the rare disease community. I however do have some comments.

In the introduction, the references to the state of the art literature are in many cases to bioarchive deposits or meeting reports (refs 2-6). As none of this has gone through peer-review, the scientific basis of this is hard to judge for any external reviewer without detailed knowledge of all aspects of the methodology.

The method seems to work well for the two example diseases cystic fibrosis and Parkinson, but I do have questions on the interpretation of the fragile X syndrome results. Complicating factor is that Fragile X syndrome is a much-studied disease and an immense number of pathways has been named involved in the disorder. This puts the numerous confirmatory examples shown here in perspective.

Moreover, in general there is a paucity of data provide. For instance, how, of all possible drugs, are only Sulindac and Ibudilast are highlighted is not made sufficiently clear. Gene expression levels are measured, but it is not clear to me at all why only a few genes are listed in the tables 1 and 2. Are these the only genes of which the expression has been altered. How many times is the up/downregulation. Why are different mouse models used for both drug screenings. Were were the mice obtained and how were these bred. What are the controls? Has a sham or other type of control be used?

The evidence for the PDE inhibitors BPN-as effective in the fragile X syndrome is indeed present. However, a reference that Ibudulast is also effective is lacking. The link with neurogenerative processes is far sought and in my opinion cannot be used as evidence. When linking the fragile X syndrome through seizures though genes and drugs, it should be realized that only 20% of fragile X patients or in fact of any intellectual disability / autism disorder suffers from seizures. Moreover, fragile-x associated seizures are relatively benign and tent to disappear as of adolescence. In other words, it is not a hallmark of the fragile X syndrome.

(ar)baclofen, here quoted as effective, trials have been a disastrous failure in human trials.

Response to Reviewers Comments

Please find a detailed description of all manuscript changes made in our response to all reviewer comments,

Reviewer 1 - Major Revisions

- My primary suggestion would be for the authors to change the focus of the manuscript to Fragile X Syndrome - discussing the predictive analytics, data that was fed into the KG that resulted in these insights, critical data gaps that exist in the rare disease world, the mechanistic explanation for predictions, followed by in vivo validation. Most of these are already in the manuscript currently but seems like an after-thought rather than the punchline which I believe it should be.*
 - We thank the reviewer for their suggestion to refocus the manuscript on the impact our evidence chain automated filtering method has on our own internal Fragile x syndrome therapeutic programme. We recognise this recommendation is made with the intention of strengthening the primary message of "real world impact in drug discovery".
 - We agree with the intent of Reviewer 1's suggestion that making these changes substantially strengthens the impact of our methodological claim. That reducing uninformative paths unlocks the potential of AI in early drug discovery. Not least by allowing readers to reproduce our efforts through the sharing of code and data. Briefly, the revised manuscript provides much clearer discussion of the predictive analytics, therapeutic hypothesis generation and in vivo validation for Fragile x syndrome.
 - The primary focus of this paper remains the automated method and its significance, but we have given more emphasis for the FragileX results in the revision and improved the logical flow.
 - [Line 146-152] The revised manuscript now benefits from a much clearer definition of objective, establishment of method, outcomes from application to Fragile x syndrome and benchmarking against Cystic fibrosis (CF) and Parkinson disease (PD) following rewording in the introduction. Given this manuscript is primarily methodological, and a significant advantage is its general applicability in all disease therapeutic areas, we did not restrict description to Fragile x syndrome alone. That said, we have slimmed down descriptions of PD and CF approved treatments mechanisms to improve clarity of messaging on their methodological use in algorithm validation.
 - [Line 21-31] Importantly, we have prefaced the paper with a short paragraph describing the challenges of rare disease drug discovery, and the problem of establishing therapeutic rationale when data is limited. We agree with the reviewer that this reworking of the story enhances the message and contribution our novel automated filtering method brings to the community.

- [Line 191] To further enhance messaging, we've reworked and included the supplementary figure within the main body of text in the Results section that shows the complete workflow of the auto-filtering model along with details of each and every phase. We believe this will increase ease of understanding of the primary learning of our research.
 - [Line 545-560] Finally, we have added a detailed description of the data that comprises Healx knowledge graph including the node and edge types and the sources and their versions. Furthermore, we provide a snapshot of Healx KG in the form of an open-source sub-graph. As mentioned above, this subgraph omits proprietary know-how, data that explicitly prohibits redistribution.
2. *The results presented for Parkinson's Disease and Cystic Fibrosis are neither novel nor mechanistically very interesting from a Knowledge Graph-driven insights perspective. These are just observations from prior knowledge that don't need predictive power. While I understand that this was presented as a 'baseline' in the absence of a gold standard, I don't think it is required to be presented as a key result which I do not think it is.*
- As mentioned in comment 1 (bullet point 4; above), we understand our description of the purpose of CF and PD results was insufficiently clear, and thereby their importance in establishing a baseline to measure the effectiveness of our method at reducing uninformative paths not evident in the results section.
 - [Line 388-390] In the revised manuscript we explicitly state that CF and PD examples are given as benchmarking data to demonstrate the likely utility of our approach and as baseline justification. Given that we make no novel therapeutic claims about CF and PD diseases, we have significantly reduced the discussion and references pointing to how these approved treatments work in these diseases. Namely, we have slimmed down descriptions of the extracted evidence chains.
 - [Line 394-398] We retain the results for both diseases to show that the method is able to predict the approved treatments and extract key evidence chains in the absence of the treatment-disease links in the input knowledge graph given this is a primary finding. Further justifying this remark, as no gold standard benchmark dataset is available to validate the filtering of evidence chains for any given disease study, we chose to establish our own baselining approach. This was achieved through manual curation of treatment links with CF and PD diseases. We then describe our observations in applying our method to these diseases compared with the known curation baseline. The comparison is a measure of effectiveness of our method, and so is described in results.

3. *Methodological innovation came across as very weak, with reimplementing of established methods like AnyBURL without any detail of what is the advancement rather than reimplementing this study delivers. The automated rule filtering method was called out as 'novel' by the authors, which seemed more logical/hierarchical than statistical. I might've missed the key point here, but if the authors wanted to project this as a novel aspect of this study, it should be a key part of the Results with a workflow diagram readers can learn from and reimplement, and not confined to Methods.*

- We agree with the comments of the Reviewer that the novelty of our method was poorly presented.
- [Line 493-495.]The revised manuscript states that our innovation is in enabling KBC models such as AnyBURL to be applied in establishing pharmacological rationale through reduction of uninformative paths without loss of biological signal. We evidence this with real world case studies such as Fragile X syndrome starting from the predictions and generation of evidence chains with application of an automatic filtering model.
- [Line 75-84, 131-137, 192-245] Here we have clearly explained the novelty of the approach, also including descriptions of related work in this area and detail how we differ. We have now included a workflow diagram in the Results section explaining each step of the pipeline with clear description, discussing the importance of the method and potential benefits derived, such as reducing time and cost involved in preclinical experiments.
- We agree with the reviewer that our innovation is with the introduction of automated filtering of evidence chains tailored to the needs of drug discovery. To our knowledge, this manuscript is the first article to address the shortfall in application of KG reasoning methods such as AnyBURL in Biotech owing to the vast quantities of uninformative output these methods bring.
- In the revised manuscript we have made the problem statement and proposed solution much more prominent. Moreover, we state clearly where our innovation may impact drug discovery.

4. *Lastly, as a reader, other than the insights discussed in the paper, there is no other usable outcome that I could take away from the paper. I would like to see a compilation of data, a snapshot of the KG that delivered these results (minus the IP restricted parts), a function/library for the automatic rule filtering approach, and/or a dictionary of rules that rare disease scientists and patient cohorts can learn from.*

- We are extremely grateful to the reviewer for highlighting this shortcoming in our manuscript. We entirely agree with the reviewer that the paper requires provision of data and code to enable readers to reproduce and extend our work. To this end we have provided details on node and edge types and their sources with the versions used. We still keep the results from the full Healx KG in the paper but for reproducibility purposes we have created a subgraph of the

Healx KG excluding nodes and edges coming from proprietary data sources and internal curation. The Significant new manuscript additions in this revision are:

- i. We provide in full the source code of our biologically aware automated filtering method built on AnyBURL.
 - ii. [Supplementary Table 5 and Supplementary Fig. 2] We provide a worked example with a minimal viable subgraph of our entire Healx KG and present results for Parkinson disease to exemplify the filtering approach.
 - iii. [561-563] We are explicit that the worked example complies with Open Source guidelines, meaning that no proprietary knowledge, commercially licensed sources, or redistributed data that would breach GPL is included.
 - iv. [568-573] We clearly state that the results of the manuscript are produced using this source code and the *full Healx KG* that accounts for any minor differences in specific evidence chain composition in the Supplementary results, yet this does not limit or distract from the main learning of our manuscript regarding reduction in uninformative paths without loss of biological signal (cf: Parkinson disease example) and general applicability of our workflow in drug discovery.
-
- We reran the evidence chains generation and auto filtering pipeline in this subgraph and demonstrate results achieved for Parkinson's disease. We are delighted to be able to share in the supplementary results the revalidation of our approach using this sub graph.
 - Briefly, we show a few of the key evidence chains extracted and the percentage of reduction achieved in the evidence chains space.
 - We provide the required code and scripts for researchers to achieve the same outcome using the bundled sub graph.
 - The dictionary of rules was already provided for CF and PD in the initial submission in the Supplementary document. However, in the revision we have provided a list of automatically generated rules from the method for Fragile X Syndrome in Supplementary Table 4.

Reviewer 1 - Minor Revisions

1. *The manuscript needs a deeper background as to why KGs are relevant in this space, and what has already been achieved. With the recent ongoing focus/scrutiny of AI, it is key to discuss where the authors believe value lies and what the key gaps are as identified by previous studies in this space from a technical (KG) as well as a scientific (repositioning in rare diseases) perspective.*

- [Line 41-56] To address this, we have included related work in the introduction discussing the use of KGs in the drug discovery space not only in drug repositioning but also in other areas like target identification. We have discussed why it's important, what has been done so far and how we differ from those. We also explain several Knowledge graph completion models from the literature and discuss how we have used a symbolic model to derive a solution for the problem of explaining the predictions.
- 2. *References are missing for key statements stated as facts*
 - We have added the highlighted missing references in the manuscript.
- 3. *The definition of 'significance' is unclear across the manuscript, from the figures it seems logical rather than statistical driven by a scoring function. I might be right or wrong, but this needs clarification with better phrasing.*
 - Validating the evidence chains is a challenging task since there is no benchmark dataset available to statistically evaluate with a score. This is why we picked two case studies CF and PD with approved treatments to showcase what could be extracted by the method when the method is unaware of the approved treatments for the disease (we manually removed them from the test data). In this regard, significance is a qualitative term based on empirical human evaluation of the useability of the remaining evidence chains post filtering. This is a current limitation that we acknowledge in text.
 - [Line 274, 497-500]. Briefly, the chains have been reviewed by drug discovery scientists and considered significant. For example, we observe good correlation between automatically extracted paths and experimentally derived transcriptional changes of selected genes and pathways (Table 1 and 2) in Fragile X, suggesting the results are significant.
- 4. *For the data sources feeding into HealxKG, important for the readers to see a dictionary of how much and what data each of these are contributing, along with the last version date. At least one data source is very out of date and needs updating.*
 - To address this, we have provided details about the Healx knowledge graph used in the paper, including the node and edge types, sources, versions used and a meta graph showing how nodes and edge types are connected. We are unable to provide information on how much of the data sources are being used to build the knowledge graph. We have shared a sub-graph of the Healx KG for reproducibility purposes excluding proprietary and internal data.
- 5. *I have also included a document with detailed comments and suggestions for the authors to consider ad address, wherever appropriate.*
 - We are especially thankful to the Reviewer for taking the time to prepare these extremely valuable recommendations that we have fully incorporated into our

manuscript. We acknowledge that making these edits vastly improves readability, accuracy and clarity of message.

- We provide in-text details of supplementary minor correction for the Editor in Reviewer 1 - Additional minor corrections or Remarks

Reviewer 2 - Major revisions

1. *Needless to say such a system is of added value to the rare disease community. I however do have some comments. In the introduction, the references to the state of the art literature are in many cases to bioarchive deposits or meeting reports (refs 2-6). As none of this has gone through peer-review, the scientific basis of this is hard to judge for any external reviewer without detailed knowledge of all aspects of the methodology.*

- [Ref 12, 13, 14] We agree with the Reviewer that peer-reviewed references are essential to establish claims of state of the art. In the revised manuscript we have replaced or extended references where possible to include peer-reviewed articles.

2. *The method seems to work well for the two example diseases cystic fibrosis and Parkinson, but I do have questions on the interpretation of the fragile X syndrome results. Complicating factor is that Fragile X syndrome is a much-studied disease and an immense number of pathways has been named involved in the disorder. This puts the numerous confirmatory examples shown here in perspective. Moreover, in general there is a paucity of data provided. For instance, how, of all possible drugs, are only Sulindac and Ibudilast are highlighted is not made sufficiently clear. Gene expression levels are measured, but it is not clear to me at all why only a few genes are listed in the tables 1 and 2. Are these the only genes of which the expression has been altered. How many times is the up/downregulation. Why are different mouse models used for both drug screenings? Where were the mice obtained and how were these bred. What are the controls? Has a sham or other type of control been used?*

- [Line 271-273] We are demonstrating in Tables 1 and 2, the observed molecular effects of predicted reasoning from evidence chains alone observed for FXS and not the entire list of genes altered in the preclinical testing. For this methodological study that would be out of scope. What we provide here is an experimental confirmation of the in-silico hypothesis of the mechanism provided by the evidence chain generation and automated filtering pipeline. That is, we show the reduction of evidence chain paths post filtering still retains mechanistically meaningful biological signals. This is reported as correlation of the top in-silico evidence chain containing genes and their observed expression derived experimentally following drug treatment in vivo.

- Sulindac and Ibudilast emerged as promising compounds in non-Knowledge Graph (KG) based drug prediction models, ranking high with high confidence as potential treatments for FXS. However, these models lacked the capability to provide a rationale for their predictions. Our unique pipeline is capable of generating evidence for such predictions, even when they originate from different models. The only provision being that the query drugs are also predicted by the AnyBURL model, as they were here.
 - [Line 699-734] We agree with the reviewer that the manuscript would benefit from more details on the preclinical experimental study. In the revised manuscript we provide more description of the experimental design including the reasons for why different mouse models were used, the number of times up/down gene dysregulation observed [Line 269-271], and the controls and the type of breeding used.
3. *The evidence for the PDE inhibitors BPN-as effective in the fragile X syndrome is indeed present. However, a reference that Ibudilast is also effective is lacking. The link with neurogenerative processes is far sought and in my opinion cannot be used as evidence. When linking the fragile X syndrome through seizures through genes and drugs, it should be realized that only 20% of fragile X patients or in fact any intellectual disability / autism disorder suffers from seizures. Moreover, fragile-x associated seizures are relatively benign and tend to disappear as of adolescence. In other words, it is not a hallmark of the fragile X syndrome.*
- We thank the reviewer for their questions and concern regarding evidence supporting the effectiveness of PDE inhibitors such as ibudilast in treating seizures in neurodevelopmental disorders such as FXS. Moreover, whether seizure is a relevant symptomology for treatment development. We realise that the manuscript would be improved by further pharmacological and clinical description, and include the additional evidence in text.
 - [Line 279-285] The validation of PDE4 as a therapeutic target in FXS has been demonstrated across several species, including FXS patients. Ibudilast does potentially have advantages over selective PDE4 inhibitors due to its broad selectivity profile against PDE3, PDE4, PDE10 and PDE11 [1]. Despite this broad selectivity profile ibudilast is more selective for PDE4 and PDE10, with IC values in the lower μM range, compared to PDE3 and PDE11 where IC values are in the 10 μM range [1]. This implies that ibudilast's PDE efficacy is likely coming from its inhibition of PDE4 and PDE10. PDE10 is a target of FMRP [2] and as a result levels are elevated in FXS. Beneficial effects of PDE10 inhibition have been demonstrated in Fmr1 KO mice by normalizing EEG recorded chirp ITPC [3]. This suggests that PDE10 inhibition reduces auditory hypersensitivity, a debilitating condition which can lead to language delays, social anxiety and stereotypy [4] all common symptoms in FXS patients. Ibudilast has the potential to improve

auditory hypersensitivity and reverse neural deficits within the basal ganglia through PDE10 modulation.

- [Line 317-321] Any phenotypic reversion is relevant and of interest to Healx's approach of drug discovery. We acknowledge that seizure incidence is low in FXS, however, several peer reviewed publications have demonstrated that by targeting one particular pathophysiological pathway can often alleviate multiple symptoms in FXS, including seizures [5,6,7,8,9,10] . This has been demonstrated in preclinical models as well as clinical studies [11]. It therefore stands to reason that a small molecule which is able to alleviate seizure incidence may improve other symptoms in FXS and therefore warrants further investigation.
4. (ar)baclofen, here quoted as effective, trials have been a disastrous failure in human trials.
- [Line 366-370] We acknowledge that Arbaclofen did not reach the primary outcome measure of improved social avoidance in FXS in a phase 3 clinical trial. However secondary measures suggested that younger patients did show some benefit and the investigators suggest that larger cohorts on higher doses were needed. The investigators came to the following conclusions: "These encouraging results suggest arbaclofen should be studied further to replicate the result. Young age, higher doses, larger cohort sizes, trial designs that minimize placebo effect, and better outcome measures covering a wide range of potential responses are among the factors that may allow success in future trials of arbaclofen and other drugs that have shown promise in FXS experimental models [12].
 - In addition to this lack of patient stratification may also have contributed to the failure of arbaclofen in FXS. Although FXS is a monogenic syndrome, patients display considerable clinical and genetic heterogeneity which manifests in a wide spectrum of behavioral phenotypes among patients [13]. This heterogeneity is thought to stem from the heterogeneous genetic background of patients as well as the existence of mosaicism of FMR1 methylation, which results in a differential expression of FMR1 across the brain [14]. It is therefore not surprising that despite several potential targets being uncovered and trialed in the clinic [15], no disease modifying therapy currently exists which is able to address the multiple symptoms in FXS patients.
 - See Appendix for additional references included in this response to reviewer comment.

Reviewer 1 - Additional minor corrections or Remarks

We provide a description of 41 changes made in the revised manuscript, listed here using Reviewer 1's reference by line number of original manuscript with cited text.

1. *Line 1 - Might be more impactful for the title to be disease specific. My personal view is that this manuscript should be FragileX focussed to deliver the most impact*
 - We thank the reviewer for their suggestion to refocus the manuscript on the impact our evidence chain automated filtering method has on our Fragile x syndrome therapeutic programme. The primary focus of this paper is still the automated method and its significance, but we have given more emphasis for the FragileX results in the revision and improved the logical flow to convey that message.
 - A more complete description of our response is provided in Reviewer 1 - Major Revisions; correction 1

2. *Line 29-30 - The list of ranked predicted drugs can differ across models even with the same drug and disease data due to differences in model logic. Agree with the statement, needs reference.*
 - We have included a reference in text to the following article that shows different evaluation results for several models trained on the same data.
 - i. Wang, Y., Ruffinelli, D., Gemulla, R., Broscheit, S., & Meilicke, C. On Evaluating Embedding Models for Knowledge Base Completion. In Proceedings of the 4th Workshop on Representation Learning for NLP (Repl4NLP-2019), 104-112, (2019)

3. *Line 35 - Grammatical error - delete shown*
 - Corrected in revised manuscript

4. *Line 44 - There are very few KBC models that can generate evidence automatically for the predictions made. Add reference.*
 - [Line 82] We have added in text the following references to related work that produces logical rules and paths as evidence for predictions including the path ranking algorithm, KBC model for path generation, Markov logic networks, RuleN, AnyBURL, Minerva and PoLo.
 - i. Lao, N., & Cohen, W. W. (2010). Relational retrieval using a combination of path-constrained random walks. Machine learning, 81, 53-67.
 - ii. Gardner, M., Talukdar, P., Kisiel, B., & Mitchell, T. (2013, October). Improving learning and inference in a large knowledge-base using latent syntactic cues. In Proceedings of the 2013 Conference on Empirical Methods in Natural Language Processing (pp. 833-838).
 - iii. Sudhahar, S., Roberts, I., & Pierleoni, A. (2019). Reasoning Over Paths via Knowledge Base Completion. EMNLP-IJCNLP 2019, 164.
 - iv. Richardson, M., & Domingos, P. (2006). Markov logic networks. Machine learning, 62, 107-136.

- v. Meilicke, C., Fink, M., Wang, Y., Ruffinelli, D., Gemulla, R., & Stuckenschmidt, H. (2018). Fine-grained evaluation of rule-and embedding-based systems for knowledge graph completion. In *The Semantic Web–ISWC 2018: 17th International Semantic Web Conference, Monterey, CA, USA, October 8–12, 2018, Proceedings, Part I 17* (pp. 3-20). Springer International Publishing.
- vi. Meilicke, C., Betz, P. & Stuckenschmidt, H. Why a naive way to combine symbolic and latent knowledge base completion works surprisingly well. *Proceedings of the 3rd Conference on Automated Knowledge Base Construction, (2021)*.
- vii. Das, R., Dhuliawala, S., Zaheer, M., Vilnis, L., Durugkar, I., Krishnamurthy, A. & McCallum, A. Go for a walk and arrive at the answer: Reasoning over paths in knowledge bases using reinforcement learning. *Proceedings of the 6th Workshop on Automated Knowledge Base Construction, (2017)*.
- viii. Meilicke, C., Chekol, M.W., Ruffinelli, D. & Stuckenschmidt, H. Anytime bottom-up rule learning for knowledge graph completion. In: *The Twenty-Eighth International Joint Conference on Artificial Intelligence (2019)*.
- ix. Liu, Y., Hildebrandt, M., Joblin, M., Ringsquandl, M., Raissouni, R. & Tresp, V. Neural multi-hop reasoning with logical rules on biomedical knowledge graphs. *ESWC 2021: The Semantic Web. 375-391, (2021)*.

5. *Lines 59-66 Whilst very relevant and useful, this paragraph can be simplified/content moved to methods with a summary included in the introduction.*

- We have simplified this paragraph and moved the rule notations with variables to the Methods section.

6. *Introduction - The introduction is well written but is lacking a key aspect of explainable KG predictions - the biological KG itself. I suggest the authors include a short paragraph on current state-of-the-art covering recent studies that discuss building a drug discovery relevant KG and/or using the same for generating a ranked list of recommendations.*

Recent studies that come to mind are:

- <https://www.biorxiv.org/content/10.1101/2022.12.20.521235v1>
- <https://www.nature.com/articles/s41467-022-29292-7>
- <https://www.nature.com/articles/s41587-021-01145-6>

- We have included a paragraph in the Introduction section covering recent studies that build KGs for drug-repurposing, target identification and clinical decision making. We have also included the references to suggested articles by the reviewer.

-
7. *Figure 1 - Unsure what KG the relationships shown in Fig 1 are derived from. If this is from a published source or a bespoke KG, it should be mentioned and cited appropriately.*
 - We have added a mention to the Healx KG that was used to produce Figure 1.
 8. *The authors do mention other studies that follow a similar approach to theirs but do not cover what other methods are out there that could potentially be beneficial in this domain.*
 - We agree and have now provided more related work in this domain which generates paths explaining a prediction using different approaches as mentioned in point 4 above.
 9. *Line 102 - Mention contradictory relationships (causal and otherwise, missing data, false data etc.)*
 - We are sorry that we do not understand what is meant by the reviewer related to the lines and failed addressing it.
 10. *Line 123 - If the paths are all 'significant', how do the authors assess/capture known vs novel relationships?*
 - Also addressed fully in Review 1 - Minor Revisions; Comment 3. Significance is a qualitative term for now but the chains have been reviewed by drug discovery scientists and considered significant for example in the FXS study where we observe that correlation between automatically extracted paths and experimentally derived transcriptional changes of selected genes and pathways was high, suggesting the results are meaningful in the context of drug discovery decision making, specifically whether to advance therapeutic hypotheses to preclinical evaluation.
 - We do not assess known vs novel in the approach. The main goal is to explain a particular therapeutic hypothesis for a given disease and a potential treatment mechanism. In doing so novel biological aspects can surface as we observed in the FXS study with respect to Sulindac and Ibudilast perturbation signatures.
 11. *Line(s) 128+ - Whilst it is a fair statement about the choice of diseases, the argument that certain diseases would generate many evidence chains seems a bit odd. Unless I've totally misunderstood the aim of this study, isn't one of the primary aims to show that this approach helps identify the high confidence and biologically/therapeutically meaningful associations helping reduce the large number of evidence chains such approaches usually result in? Seems a bit counter-intuitive to say that the disease choice is to reduce numbers in the first place. I have no issues with the authors' choice of diseases, just the argument needs to be rephrased.*
 - We agree with the reviewer's point here and acknowledge that the criteria for selection of the baseline diseases was insufficiently clear.

- Our disease choice in this instance was to select workable diseases that require comprehensive manual review used for establishing empirical performance. This being very different to a de novo discovery programme where no prior constraints would exist, as was the case for FXS.
 - We have rephrased our argument for not choosing complex diseases [Line 152-157]
 - Briefly, we avoided complex diseases such as cancers and auto-immune conditions since validating the output against known information for these diseases would involve too many genes and pathways and likely form an overly complex case study of our automatic evidence chains filtering baselining approach.
12. *Line 129 - Define 'ancestor' relationship. Is this derived from an established disease ontology (which it looks like it is), then it needs to be specified?*
- In the revised manuscript we list the sources of the relationship from disease ontologies MONDO and Orphanet and provide references to it in text.
13. *Line 130 - Even if the disease focus is not cancer/auto-immune disorders, this space had quite a few interesting publications in the last couple of years. Suggest citing them to give the readers a more comprehensive citation list.*
- We have included a few references to studies using Knowledge graph approaches in cancer and auto-immune conditions in the revision.
 - i. Gogleva, A., Polychronopoulos, D., Pfeifer, M., Poroshin, V., Ughetto, M., Martin, M. J., ... & Bulusu, K. C. (2022). Knowledge graph-based recommendation framework identifies drivers of resistance in EGFR mutant non-small cell lung cancer. *Nature communications*, 13(1), 1667.
 - ii. Daowd, A., Abidi, S., & Abidi, S. S. R. A Knowledge Graph Completion Method Applied to Literature-Based Discovery for Predicting Missing Links Targeting Cancer Drug Repurposing. In *Proceedings of International Conference on Artificial Intelligence in Medicine*, Springer International Publishing. 24-34 (2022)
 - iii. Pu, Y., Beck, D., & Verspoor, K. Graph embedding-based link prediction for literature-based discovery in Alzheimer's Disease. *Journal of Biomedical Informatics*, 145, 104464 (2023)
14. *Line 133 - Also, just in the last year, some interesting PD KG studies were published that looked at predicting novel targets and early markers of disease that could potentially suggest therapeutic interventions. Important to mention these. One I just came across: <https://bmcbioinformatics.biomedcentral.com/articles/10.1186/s12859-021-04530-9>*
- We have added references to studies using knowledge graph approaches in PD including the reference mentioned by the reviewer.

15. Line 140 - given this is the first account of a systematic evaluation of this type What do the authors mean by 'this type'? If it is evidence chain generation, then not the first as there are other published PD KGs. If they mean methodological 'first', needs clarification if it is the scalability or the underlying mathematics that is novel.

- Following a thorough literature survey, we are unaware of any prior study that evaluated evidence chain paths utility and performance specifically with the intention of informing on treatment mechanisms applied in real world drug discovery.
- Our literature surveys suggest there are no benchmark datasets available for this purpose and creating such a data set would be time consuming to the point of infeasibility. 'This type' means the actual evidence chains generated using a KBC model, path generation and the application of a filtering model that extracts biologically meaningful evidence chains.
- Empirically, we evaluated our pipeline in different aspects, first with a curated set of rules and then by comparison with the auto-filtering model's output against the curated baseline rule set. In FXS we provided an indication of the utility of this approach to inform on drug mechanisms in our own internal drug discovery pipeline.

16. Line 144 - As the authors noted previously, 'relevance' does not equate to 'causal' or 'therapeutic relevance'. Whilst there is immense value in expert valuation of KG recommendations, assigning confidence/weight would be more important than indicating relevance.

- We agree with the reviewer that assigning a confidence or weight to the evidence chains is important. Although there is a weight associated with the rules given by AnyBURL it does not equate to therapeutic relevance in our experience. We mention in the 'Discussion' text that methods to attach scores will be explored in a future work, thus quantitative ways to express confidence in evidence chains.

17. Line 146 - How independently were these rules generated? Were they given any guidance/definitions to ensure comparability at the same time as being unbiased?

- We made the entire set of rules from AnyBURL available for review and requested curators to assess the general biological relevance of these rules in the context of any disease, not just the disease of interest. This approach eliminated bias toward a specific disease and ensured comparability in the curation process. This point has been made clear in the manuscript [Line 180-182]. We checked whether the automated filtering method can retain all the curated rules before generating evidence.

18. Line 151 - *AnyBURL is a published and well-used methodology. Have the authors built upon this in any way to highlight a technological advancement?*

- Our innovative contribution is the development of an automated filtering workflow that reduces prediction noise of the *AnyBURL* model. The impact is a significant reduction in the number of generated paths without loss of biological signal. This allows the *AnyBURL* model to support understanding of the therapeutic basis of a treatment when used in conjunction with the automated filtering workflow. We have not made any changes to the *AnyBURL* algorithm directly.

19. Line 166 - *'we intend to establish the performance and validate the utility of our approach by rediscovering a set of known treatment relationships along with valid evidence chains that explain those treatments': is this different to the lack of 'gold standard' the authors mentioned in the previous paragraph?*

- It is true that due to the lack of gold standard for evidence chains between a disease and a drug we had to use other ways of validating our approach. The case studies presented in Cystic fibrosis and Parkinson's disease is a baseline justification of the approach and confirms that we can predict those treatments in the absence of the treatment links in the graph. We can produce valid evidence chains that automatically extracts key genes and pathways involved in understanding the mechanism of action of the drug.

20. Line 174 - *How many independent pieces of evidence support each one of the 45 PD-treatment relationships? I'm assuming it would be quite high given these are 'known' relationships. If so, why is it an interesting finding that 44/45 were recovered?*

- In total for the 44 treatments in PD, we produced ~ 34,000 paths from Healx KG as shown in Table 3 in the manuscript. We had removed all the direct links between the treatments and PD before running *AnyBURL* and the pipeline on the graph. The method was able to rediscover/predict 44 treatment relationships and produce explainable paths for those 44 treatments. The focus of this work is still on producing explainable paths and not improving the predictive performance.

21. Line 178 - *Typo – CFTR.*

- Corrected in revised manuscript.

22. Fig 2 - *Fig 2 is a very neat summary of all evidence chains linking established treatments/interventions to CF and PD. However, as per an earlier comment, these are also the most significant/widely reported evidence chains too. Why is it a 'result' that the most reported relationships are being picked up by the 'novel' method? Even the less significant interventions in (b) are also well-reported.*

- We now realise the purpose of CF and PD was unclear. In the revised manuscript we explicitly state that CF and PD examples are given as benchmarking data to demonstrate the likely utility of our approach and as baseline justification or validation. We keep these results for both diseases to show that the method is able to predict the approved treatments and extract key evidence chains *in the absence of the treatment links between the approved treatments and the diseases in the input knowledge graph*.
- The main goal is to explain a particular therapeutic hypothesis for a given disease and a potential treatment. In doing so, novel biological aspects can surface. It might be less interesting to the disease of interest, but still provide useful insight for further experimentation in some cases.

23. *Line 197 - I'm still struggling with what's prediction vs observation in these results. All compounds discussed are well-established and would be part of a direct edge in the underlying KG. If not, the authors should comment on why not.*

- We agree with the reviewer that the original manuscript was unclear about how the baselining experiment was performed. We have considerably reworked this part of the text to make it clearer that the direct links between the treatment drugs and the diseases CF and PD were removed prior to training the model on the graph [Lines 398-400]
- The model predicted those 'known treatments' in the absence of direct links and produced meaningful evidence chains for those treatments.

24. *Line 226 - How many evidence chains were generated? How were they triaged into find the 'most significant'?*

- For Parkinson's disease the following number of evidence chains were generated for the drugs shown in Figure 4b: Rivastigmine:424, Cabergoline:1296, Entacapone:347. In total for all the 44 approved treatments the method extracted 34,738 chains, as shown in Table 3. The evidence chains shown in the pictures were chosen to represent a few examples from the full list to show how the evidence chains capture key results that are already known. We did not choose them by significance score as we already noted that scores do not convey biological significance well.
- In rare diseases without approved or known treatments, the total number of chains extracted following automated filtering for any drug is anticipated to be relatively few compared to PD and CF. In fact we have seen this already in the FXS study, making it feasible for routine review by drug discovery experts.

25. *Line 226-237 - This section reads well, but I still have the same question as with CF - observation vs prediction? PD is not my area of expertise in any way, but a quick search tells me Cabergoline has been associated with LDOPA-induced complications in PD*

patients for over 25yrs. Mechanism-of-action also seems to be very well studied. Why is this a primary result of the study?

- As mentioned above we have shown this as a result since it validates the output of the approach. The model predicted Cabergoline as a treatment in the absence of the direct link between Cabergoline and PD in the graph and was able to construct a path that explained the therapeutic basis of the drug in PD
- Further, from a drug discovery perspective effective but unsafe treatment may serve as a springboard to discover alternate active drug classes and potentially new targetable disease biology. For example, via scaffold hopping. This is relevant to mention as the primary aim of the automated filtering approach is to rapidly surface biologically meaningful pharmacological rationale with the potential for novelty.

26. *Table 2 - 'Genes and pathways found in evidence chains that have an effect in preclinical experiments in the Fmr1 KO mouse model for Sulindac.' Are these the only statistically significant genes and pathways found in the evidence chains, or a selected few have been presented?*

- For Sulindac, after filtering we were able to find 249 evidence chains and there were many genes and pathways reported. Out of those we were able to statistically confirm the significance of the three genes and pathways mentioned in Table 2.

27. *Line 315 -Ibuprofen has been reported to have anti-inflammatory properties through cAMP signalling, TLR4 inhibition, and has been shown to protect against reactive oxygen species, a common precursor to inflammation. Needs reference*

- We have added the following references:
 - i. Schepers, M., Tiane, A., Paes, D., Sanchez, S., Rombaut, B., Piccart, E., ... & Vanmierlo, T. Targeting phosphodiesterases—towards a tailor-made approach in multiple sclerosis treatment. *Frontiers in Immunology*, 10, 1727 (2019)
 - ii. Ledebroer, A., Hutchinson, M. R., Watkins, L. R., & Johnson, K. W. Ibuprofen (AV-411) a new class therapeutic candidate for neuropathic pain and opioid withdrawal syndromes. *Expert opinion on investigational drugs*, 16(7), 935-950 (2007)
 - iii. Hutchinson, M. R., Zhang, Y., Shridhar, M., Evans, J. H., Buchanan, M. M., Zhao, T. X., ... & Watkins, L. R. Evidence that opioids may have toll-like receptor 4 and MD-2 effects. *Brain, behavior, and immunity*, 24(1), 83-95 (2010)

-
28. *Line 380 - Have overall metrics of the KG (not just for the diseases of interest) been included? At 3-hop, pretty much the entire KG could be traversed. What is the size and density of the KG? Add cross-ref to the table here.*
- We apologise there has been an error and its only 2-hop paths that were computed and this is the maximum length of an evidence chain. Increasing the number of hops lead to uninformative chains according to experts. In the revised manuscript we have included the total number of 2-hop paths computed with Healx KG and cross-referenced Table 3.
29. *Section starting 379 - Why isn't the 'novel' auto-filtering one of the 'results'?*
- We agree with the reviewer and have included a section named Evidence chains generation workflow in the Results section showing and explaining the entire pipeline.
30. *Line 425 - Agree that future work is key to establish robustness of the method presented, but rule confidence should be discussed in this paper. It is critical to assign any confidence to the predictions.*
- The predictions as given by AnyBURL have a confidence score and a ranking, but in this manuscript we are not discussing the prediction performance and therefore did not include the details on scores or ranks of predictions. We agree that rule and path confidence is critical, and we have discussed that in the revised manuscript [Line 512-522]
31. *Methods - I expect the first part of the Methods section to be a description of the Knowledge Graph, how it was constructed, what data was fed into it, what thresholds and QC were applied prior to Kg construction, size of the KG etc.? Currently, I can only imagine what the triples are from the figures presented in this manuscript. Without this, it is difficult to assess the performance of any downstream analytics.*
- We completely agree with this and thank the reviewer for highlighting it. To this end we are not able to share the complete Healx KG for confidentiality reasons, but we have provided details on node and edge types and their sources with the versions used along with a meta graph showing how nodes and edges are connected in the Methods section under 'Data'.
 - We have kept the results from the Healx KG in the paper, but for reproducibility and data sharing purposes we have created a subgraph of the Healx KG, excluding nodes and edges from proprietary data sources and internal curation. We reran the evidence chain generation and auto filtering pipeline in this subgraph and demonstrate results achieved for Parkinson's disease. We are delighted to be able to share in the supplementary results the successful revalidation of our approach using this sub graph. We show some of the key

evidence chains extracted and the percentage of reduction achieved in the evidence chains for 34 predicted treatments in Parkinson's disease. The data limits in the sub-graph is the reason why only 34 were predicted out of the 45 approved treatments.

32. Line 463+ -Several approaches have been proposed in the past for KBC that learn a continuous vector space representation for entities and relations. These methods include translational models using distance-based scoring (TransE [65], TransH [66], RotatE [67]), semantic matching models (RESCAL [68], DistMULT [69], Complex [70]), Graph convolutional networks (GCN [71], R-GCN [72]), Attention networks (GAT [73]) and context-based encoding approaches (KG-BERT [74]). Yet these models cannot produce a rationale for the predictions. Alternatively, there are comparatively few models such as AnyBURL that target the symbolic space capable of also producing a set of learnt logical rules for each prediction. This section reads like an introduction rather than Methods. Suggest the authors include content here that have been applied/developed as part of the study/interpretation rather than a background.

○ We agree and have now moved this to the Introduction section.

33. Lines 504-509 - Worth noting that rules 3,4,5 also involve many independent degrees of freedom with their own assumptions. A high confidence $A \rightarrow B$ relationship, and another high confidence $B \rightarrow C$ relationship does not necessarily mean $A \rightarrow C$ is high confidence, unless all feedback mechanisms are captured within the same ruleset definition. Whilst I agree with the wider biological context that 3,4,5 can capture, worth a sentence in Discussion acknowledging the independent factors at play here.

○ We agree and have included this in the revised manuscript under Discussion [Line 516-518].

34. Line 536 - Rules containing more than one disease_disease_ancestor or disease_disease_descendant relationships will be ignored. Will this miss out related but rare disease subtypes?

○ It depends on the structure of the input graph and how the sub types are categorised. In our case with the Healx KG, we expected one link of the sub-type with the actual rare disease, so the essential information is available in the evidence chains and we can alter the filter to accommodate more subtypes if we want to be specific. Another reason was to avoid too many complex evidence chains of more than 1 ancestor or descendant relations creating noise in the process of understanding the rationale. Therefore, depending on the input graph and intention, these filters could be altered if more subtypes have to be analysed.

-
35. *Line 544+ - 'The underlying rationale for this filter is firstly an evidence chain with more than one ancestor or descendant relationship becomes uninformative when trying to understand a potential treatment like explained in Figure 1c'. Agree that it is key when 'explaining' a relationship, but not when 'predicting' as novelty is more important there. Do the authors tweak the automatic filtering process based on which of the two (explain vs predict) is their primary question?*
- We do not differentiate explicitly between explain vs predict. Although the primary goal is to explain a particular therapeutic hypothesis for a given disease and a potential treatment, we do not adjust the automatic filtering to do this. In this process novel biological aspects can surface. It might be less interesting to the disease of interest as shown in Figure 1c, but still provide useful insight for further experimentation in some cases, which could contribute to novelty.
36. *HealxKG - Fully appreciate that the Healx KG is IP of the company, but unless this was presented/published previously, important for the readers to have an idea as to what it contains, sources of the data (internal/external), any QCs applied etc. Obviously to the extent IP restrictions allow the authors to do so. And is there a non-confidential version of the KG that the authors can share for the readers' benefit?*
- We thank the reviewer for understanding the IP restrictions we have in sharing the Healx KG. To address it, we have provided details on node and edge types along with the sources and their versions from which it was built, including a meta graph. We have shared with the revision a non-confidential version of the Healx KG which is a sub-graph of the full KG and repeated experiments in this data for Parkinson's disease. We have also shared a github repository with the source code to complete the experiments. The experimental results from this sub-graph are included in the Supplementary document enabling readers to reproduce the results.
37. *Line 561 - Was manual curation also performed by domain experts in the disease areas described in the manuscript? Otherwise it amounts to cross checking literature and not expert curation.*
- Yes, we have not made this clear in the manuscript. In the revision we mention clearly that experts performed curation related to the disease areas mentioned in the manuscript [Line 225]
38. *Line 568 - This might've been covered somewhere in the main text but how were 'pathways' defined? Why was this choice of source made given there are many well-established databases available?*
- The pathways were sourced from KEGG, Reactome and Wikipathways and are defined as a series of interconnected biochemical reactions that occur within a

cell or organism, leading to a specific biological outcome. In our representation, we simplify their representation as an ontological feature collecting genes that act together for the purposes of cell process regulation.

- We had explained this and included it under “Biological relevance gathering” in the “Evidence chains workflow” section in Results [Line 235-237]. We used GDAs curated in the previous step to perform pathways enrichment analysis using Fisher’s Exact Test method to map sets of genes to pathway terms. Furthermore, we have applied Benjamini-Hochberg correction to account for multiple hypotheses testing and then used an adjusted p-value threshold of 0.01 to extract the most statistically significant pathways.

39. *Line 590+ - Does deductive reasoning always prioritise ‘treats’ over ‘in trial for’? If yes, this step is not contributing to any new insights. If not, then the authors should include a simple chart showing the overlap and complement numbers across all CF predictions.*

- Deductive reasoning aims to build paths that do exist in the graph but are not automatically extracted in the evidence chain generation process after the rule-based and significant path filters. It is correct that ‘treats’ is always prioritised over ‘in trial for’ or ‘in orphan designation for’ relations, even in the significant path filter. Even so, there could be redundant paths between the same nodes as shown in Figure 6. Deductive reasoning further eliminates redundant paths that were not identified in previous steps.

40. *Table 4 - Useful to include entity and relationship types, will help guide the readers in building their own KGs.*

- We have included the edge types and sources in the Supplementary Tables 1 and 2.

41. *Line 647 - SIDER is a nice resource but very out of date. Highly recommend the authors re-run their workflow with a more recent version of another AE database, or at least show that this key gap is bridged by AE edges from other data sources listed in this section.*

- Internally we have current, high quality COMPOUND_causes_PHENOTYPE side-effect information from NLP models. For confidentiality reasons we are unable to include this information in our published subgraph, so we choose to include the SIDER resource instead.

Reviewer 2 - References

1. Gibson, L. C. D. et al. The inhibitory profile of Ibudilast against the human phosphodiesterase enzyme family. *Eur. J. Pharmacol* 538, 39-42 (2006).
2. Maurin, T. et al. HITS-CLIP in various brain areas reveals new targets and new modalities of RNA binding by fragile X mental retardation protein. *Nucleic Acids Res* 46(12), 6344-6355 (2018).
3. Jonak, C. R. et al. The PDE10A Inhibitor TAK-063 Reverses Sound-Evoked EEG Abnormalities in a Mouse Model of Fragile X Syndrome. *Neurotherapeutics* 18, 1175-1187 (2021).
4. Razak, K. A., Binder, D. K. & Ethell, I. M. Neural Correlates of Auditory Hypersensitivity in Fragile X Syndrome. *Frontiers Psychiatry* 12, 720-752 (2021).
5. Berry-Kravis, E., Knox, A., & Hervey, C. Targeted treatments for fragile X syndrome. *Journal of Neurodevelopmental Disorders*, 3(3), 193-210 (2011).
6. McBride, S. M., Choi, C. H., Wang, Y., Liebelt, D., Braunstein, E., Ferreira, D., ... & Jongens, T. A. Pharmacological rescue of synaptic plasticity, courtship behavior, and mushroom body defects in a *Drosophila* model of fragile X syndrome. *Neuron*, 45(5), 753-764 (2005)
7. Yan, Q. J., Rammal, M., Tranfaglia, M., & Bauchwitz, R. P. Suppression of two major Fragile X Syndrome mouse model phenotypes by the mGluR5 antagonist MPEP. *Neuropharmacology*, 49(7), 1053-1066 (2005).
8. Dölen, G., Osterweil, E., Rao, B. S., Smith, G. B., Auerbach, B. D., Chattarji, S., & Bear, M. F. Correction of fragile X syndrome in mice. *Neuron*, 56(6), 955-962 (2007).
9. Choi, C. H. et al. Age-dependent cognitive impairment in a *Drosophila* Fragile X model and its pharmacological rescue. *Biogerontology* 11, 347-362 (2010).
10. Choi, C. H. et al. Pharmacological reversal of synaptic plasticity deficits in the mouse model of Fragile X syndrome by group II mGluR antagonist or lithium treatment. *Brain Res.* 1380, 106-119 (2011).
11. Berry-Kravis, E., Knox, A. & Hervey, C. Targeted treatments for fragile X syndrome. *J. Neurodev. Disord.* 3, 193-210 (2011).
12. Berry-Kravis, E. et al. Arbaclofen in fragile X syndrome: results of phase 3 trials. *J. Neurodev. Disord.* 9, 3 (2017).
13. Verdura, E. et al. Heterogeneity in Fragile X Syndrome Highlights the Need for Precision Medicine-Based Treatments. *Frontiers Psychiatry* 12, 722378 (2021).
14. Nolin, S. L., Glicksman, A., Houck, G. E., Brown, W. T. & Dobkin, C. S. Mosaicism in fragile X affected males. *Am. J. Med. Genet.* 51, 509-512 (1994).
15. Jacquemont, S. et al. The challenges of clinical trials in fragile X syndrome. *Psychopharmacology* 231, 1237-1250 (2014)

Reviewers' Comments:

Reviewer #1:

Remarks to the Author:

I thank the authors of this study for their sincere consideration of all my feedback and suggestions. I am satisfied that the detailed changes to both the text and flow of the manuscript the authors have made to the latest manuscript addresses all my major concerns.

Reviewer #2:

Remarks to the Author:

The authors have adapted the manuscript according to the suggestions made by the referees.